# A Data-Driven Prism: Multi-View Source Separation with Diffusion Model Priors

**Sebastian Wagner-Carena**[1,2,*]
swagnercarena@flatironinstitute.org

**Aizhan Akhmetzhanova**[1,3,*]
aakhmetzhanova@g.harvard.edu

**Sydney Erickson**[4]
sydney3@stanford.edu

[1] Center for Computational Astrophysics, Flatiron Institute
[2] Center for Data Science, New York University
[3] Department of Physics, Harvard University
[4] Department of Physics, Stanford University

## Abstract

A common challenge in the natural sciences is to disentangle distinct, unknown sources from observations. Examples of this source separation task include deblending galaxies in a crowded field, distinguishing the activity of individual neurons from overlapping signals, and separating seismic events from an ambient background. Traditional analyses often rely on simplified source models that fail to accurately reproduce the data. Recent advances have shown that diffusion models can directly learn complex prior distributions from noisy, incomplete data. In this work, we show that diffusion models can solve the source separation problem without explicit assumptions about the source. Our method relies only on multiple views, or the property that different sets of observations contain different linear transformations of the unknown sources. We show that our method succeeds even when no source is individually observed and the observations are noisy, incomplete, and vary in resolution. The learned diffusion models enable us to sample from the source priors, evaluate the probability of candidate sources, and draw from the joint posterior of the source distribution given an observation. We demonstrate the effectiveness of our method on a range of synthetic problems as well as real-world galaxy observations.

## 1 Introduction

For scientific data, pristine, isolated observations are rare: images of galaxies come blended with other luminous sources [1–3], electrodes measuring brain activity sum multiple neurons [4–6], and seismometers registering earthquakes contend with a constant seismic background [7, 8]. Additionally, the observations are often incomplete and collected by a heterogeneous set of instruments, each with unique resolutions. The corrupted data is rarely directly usable. Instead, leveraging these datasets for scientific discovery requires solving a source separation problem to either learn the unknown source prior [9, 10] or constrain the posteriors for individual sources given an observation [1, 6, 7]. In this work, we address the general challenge of multi-view source separation (MVSS).

Most source separation methods, including ICA-based methods [11–13], non-negative matrix factorization methods [14–16], and template-fitting methods [17–19], require strong prior assumptions

---

*Equal contribution.

about the sources. Similarly, most deep-learning-based methods require access to samples from the source priors to generate training sets [20–25]. When the source distributions are not well-understood, this poses a degeneracy: isolating and measuring the source signals requires a source prior, but constraining the source prior requires isolated measurements of the sources.

Alternatively, some source separation methods assume a known mixing process and thereby relax the need for a source prior [26–29]. To break the degeneracies between the sources, these methods rely on distinct collections of observations, or views, with each view offering a different linear mixture of the underlying sources. These works focus on contrastive datasets, where the goal is to separate a signal that is enriched in a target view compared to a background view. While relevant for a number of scientific datasets, these source separation methods are either limited in their expressivity [28, 29] or are not designed for incomplete data [26]. Additionally, the contrastive assumption fails in domains where no source is ever individually measured.

Recent work has shown that score-based diffusion models [30] can serve as expressive Bayesian priors. Notably, once a diffusion model prior is trained, it enables effective posterior sampling for Bayesian inverse problems [31–39]. In the setting of noisy, incomplete observations, embedding diffusion models within an expectation-maximization framework can be used to learn an empirical prior [40]. In this work, we extend the use of diffusion model priors to MVSS. By leveraging the ability to sample joint diffusion posteriors over independent sources, our method directly learns a prior for each source. The main contributions of our method are:

**Generalist method for multi-view source separation:** Our method is designed for any MVSS problem that is identifiable and linear. We show experimentally that our method works even when the data is incomplete, noisy, and varies in dimensionality. Additionally, our method does not require contrastive examples and succeeds even if every source is present in every observation.

**Source priors and posteriors:** Our method results in independent diffusion models for each source. This affords all of the sampling and probability density evaluation benefits of diffusion models.

**State-of-the-art (SOTA) performance:** Our method outperforms existing methods on the contrastive MVSS problem despite having a more generalist framework.

## 2  Problem Statement

Consider a noisy observation $\mathbf{y}^\alpha$ of view $\alpha \in \{1, \ldots, N_{\text{views}}\}$ which is composed of a linear mixture of distinct sources $\mathbf{x}^\beta$ with $\beta \in \{1, \ldots, N_s\}$. The exact mixture of each source is given by a matrix $A_{i_\alpha}^{\alpha\beta}$ that depends on the view, $\alpha$, the source, $\beta$, and the specific sample, $i_\alpha$. This model can be formalized as:

$$\mathbf{y}_{i_\alpha}^\alpha = \left( \sum_{\beta=1}^{N_s} \mathbf{A}_{i_\alpha}^{\alpha\beta} \mathbf{x}_{i_\alpha}^\beta \right) + \eta_{i_\alpha}^\alpha, \tag{1}$$

where $\eta_{i_\alpha}^\alpha \sim \mathcal{N}(0, \mathbf{\Sigma}_{i_\alpha}^\alpha)$. The $\alpha$ subscript on the sample index $i$ highlights that sample indices between views are unrelated. Importantly, source draws are not shared between views.

**Goal**: Given samples of noisy observations in each view, we aim to infer the individual prior distributions $p(\mathbf{x}^\beta)$ of each source $\{\mathbf{x}^\beta\}_{\beta=1}^{N_s}$. Access to these source priors then allows us to perform source separation by sampling from the joint posterior $p(\{\mathbf{x}^\beta\}|\mathbf{y}_{i_\alpha}^\alpha, \mathbf{A}_{i_\alpha}^{\alpha\beta})$.

**Dimensionality:** Unlike traditional source separation, the dimensionality of the observation is determined by the specific view: $\mathbf{y}^\alpha, \eta^\alpha \in \mathbb{R}^{d_\alpha}$ and $\mathbf{\Sigma}^\alpha \in \mathbb{R}^{d_\alpha \times d_\alpha}$. Similarly, the source dimensionality can vary between sources, $\mathbf{x}^\beta \in \mathbb{R}^{d_\beta}$, leading to a mixing matrix whose dimensionality is determined by the view and source, $\mathbf{A}^{\alpha\beta} \in \mathbb{R}^{d_\alpha \times d_\beta}$. No assumption is placed on the relative magnitude of the $d_\alpha$ and $d_\beta$ values, although for many applications $d_\beta \geq d_\alpha \, \forall \, \alpha, \beta$.

**Source Independence:** We assume that each source is conditionally independent of the other sources, allowing us to factorize the prior distribution: $p(\{\mathbf{x}^\beta\}_{\beta=1}^{N_s}) = \prod_{\beta=1}^{N_s} p(\mathbf{x}^\beta)$. Similarly, we assume independence between the sources and mixing matrices: $p(\mathbf{x}^\beta|\mathbf{A}_i^{\alpha\beta'}) = p(\mathbf{x}^\beta) \, \forall \, \beta, \beta', \alpha$.

**Mixing Matrix and Incomplete Data:** In contrast to *blind* source separation, we will assume that the mixing matrices, $\mathbf{A}_{i_\alpha}^{\alpha\beta}$, are known. However, while the dimensionality of the mixing matrices are fixed for each view and source, the specific matrix can differ between samples $i_\alpha$. For the purposes of this work, we will consider any non-invertible linear transformation to generate *incomplete* data.

**Identifiability**: Not all choices of mixing matrices and dimensionalities will lead to a unique solution for the prior distributions. When a unique solution does not exist, any MVSS method will converge to a set of source distributions that accurately describe the data but may not match the true distributions. Therefore, we will assume identifiability throughout this work.

## 3 Related Works

**Multi-view Source Separation:** Research on MVSS problems has focused on the contrastive setting. In this setting, there are two views, the background view that contains only the background source and the target view that contains both the background source and the target source:

$$\mathbf{y}_i^{\text{bkg}} = \mathbf{x}_i^{\text{bkg}} + \eta_i^{\text{bkg}}; \quad \mathbf{y}_j^{\text{targ}} = \mathbf{x}_j^{\text{bkg}} + \mathbf{x}_j^{\text{targ}} + \eta_j^{\text{targ}}. \tag{2}$$

Contrastive latent variable models (CLVM; [28]) define two sets of latent distributions in a lower-dimensional space, one for the background source and one for the target source. The latent variables are mapped to the observed space either with a linear model (CLVM - Linear) or through a non-linear transformation parameterized by a neural network (CLVM - VAE). The parameters controlling the transformation from the low-dimensional latent space to the observation space are optimized through an expectation-maximization or variational inference approach. The CLVM method can generate posterior and prior samples for both sources, and it can be adapted to incomplete data.

Contrastive principal component analysis (CPCA; [27]), and its probabilistic extension (PCPCA; [29]), attempt to find vectors that maximize the variance in the target view without explaining the variance in the background view. They do so by introducing a pseudo-data covariance matrix $C = C_{\text{targ}} - \gamma C_{\text{bkg}}$, with $\gamma$ a tunable hyperparameter. PCPCA can only sample from the target source posterior and prior, but it can be adapted to incomplete data.

Contrastive variational autoencoder models (CVAE; [26]) are an alternative formulation of the CLVM-VAE method[2]. In contrast to CLVM-VAE, CVAE uses two encoders shared across views: one produces background latents and the other produces target latents. The concatenated latents are fed to a shared decoder, with the target latents multiplied by zero for the background decoding task. In the original formulation, the CVAE model never outputs the target source during training, only the target observation. This allows for non-linear mixing of target and source, but makes extending the model to incomplete data challenging when $\mathbf{x}_j^{\text{bkg}}$ and $\mathbf{x}_j^{\text{targ}}$ do not share a mixing matrix.

**Diffusion Models:** Diffusion models [30, 41–44] seek to reverse a known corruption process in order to be able to generate samples from a target distribution. In the continuous-time framing, samples from the target distribution, $x_0 \sim p(x_0)$, are corrupted through a diffusion process governed by the stochastic equation:

$$\mathrm{d}\mathbf{x}_t = f(\mathbf{x}_t, t)\mathrm{d}t + g(t)\mathrm{d}\mathbf{w}_t, \tag{3}$$

where $f(\mathbf{x}_t, t)$ is known as the drift coefficient, $g(t)$ is known as the diffusion coefficient, and $\mathbf{w}_t$ is generated through a standard Wiener process. The time coefficient $t$ ranges from 0 to 1, with $\mathbf{x}_0$ being the original samples and $\mathbf{x}_1$ being the fully diffused samples. This induces a conditional distribution of the form $p(\mathbf{x}_t|\mathbf{x}_0) = \mathcal{N}(\mathbf{x}_t|\alpha_t\mathbf{x}_0, \mathbf{\Sigma}_t)$, where $\alpha_t$ and $\mathbf{\Sigma}_t$ can be derived from our drift and diffusion coefficients [45]. The forward stochastic differential equation (SDE) has a corresponding reverse SDE [46]:

$$\mathrm{d}\mathbf{x}_t = \left[f(\mathbf{x}_t, t) - g(t)^2 \nabla_{\mathbf{x}_t} \log p(\mathbf{x}_t)\right] \mathrm{d}t + g(t)\mathrm{d}\bar{\mathbf{w}}_t, \tag{4}$$

where $\bar{\mathbf{w}}$ is the standard Wiener process with time reversed. This reverse SDE evolves a sample from the fully diffused distribution $p(\mathbf{x}_1)$ back to the original data distribution $p(\mathbf{x}_0)$. Equation 4 requires access to the score function, $\nabla_{\mathbf{x}_t} \log p(\mathbf{x}_t)$, which is approximated by a neural network

---

[2]We arbitrarily use CLVM-VAE versus CVAE to distinguish between Severson et al. [28] and Abid et al. [26].

trained via score matching on samples from the forward diffusion process [30, 47, 48]. Training and sampling from a diffusion model requires selecting an SDE parameterization [30, 42, 44, 49], a score matching objective [30, 42, 43, 45], and a sampling method for the reverse SDE [30, 42, 44, 45].

We adopt the variance exploding parameterization for the SDE [43], the denoiser parameterization from Karras et al. [45] for the score matching approach, and the predictor-corrector (PC) algorithm as our sampling method [30]. The denoiser parameterization approximates $\mathbb{E}[\mathbf{x}_0|\mathbf{x}_t]$ by minimizing the objective:

$$\mathcal{L}(\theta) = \mathbb{E}_{p(\mathbf{x}_t|\mathbf{x}_0)} \left[ \lambda(t) \| d_\theta(\mathbf{x}_t, t) - \mathbf{x}_0 \|_2^2 \right] . \tag{5}$$

Here, $d_\theta(\mathbf{x}_t, t)$ is our denoiser model with parameters $\theta$. We use a loss weighting term, $\lambda(t)$, to ensure all time steps are equally prioritized. Note the denoiser returns the expectation value, which can be directly converted to the score via Tweedie's formula [50].

**Posterior Sampling:** Once trained, a diffusion model can be used as a Bayesian prior for conditional posterior sampling [30]. Specifically, the score function in the reverse SDE is replaced by the posterior score function:

$$\nabla_{\mathbf{x}_t} \log p(\mathbf{x}_t|\mathbf{y}) = \nabla_{\mathbf{x}_t} \log p(\mathbf{x}_t) + \nabla_{\mathbf{x}_t} \log p(\mathbf{y}|\mathbf{x}_t), \tag{6}$$

where $\mathbf{y}$ is our observation, and $\nabla_{\mathbf{x}_t} \log p(\mathbf{y}|\mathbf{x}_t)$ is the score of the likelihood. The prior score in Equation 6 is given by the trained diffusion model, but evaluating the likelihood score with respect to an arbitrary $t$ requires solving:

$$p(\mathbf{y}|\mathbf{x}_t) = \int p(\mathbf{y}|\mathbf{x}_0) p(\mathbf{x}_0|\mathbf{x}_t) d\mathbf{x}_0. \tag{7}$$

Many methods have been proposed for evaluating the conditional score [51]. Of particular interest are methods that propose an approximation to the right-most conditional distribution in Equation 7 [34, 36, 38–40]. In general, these methods use a multivariate Gaussian approximation:

$$p(\mathbf{x}_0|\mathbf{x}_t) \approx \mathcal{N} \left( \mathbf{x}_0 | \mathbb{E}[\mathbf{x}_0|\mathbf{x}_t], \mathbb{V}[\mathbf{x}_0|\mathbf{x}_t] \right) . \tag{8}$$

When the observation function is defined by the linear matrix $\mathbf{A}$ and the likelihood is Gaussian, $p(\mathbf{y}|\mathbf{x}_0) = \mathcal{N}(\mathbf{y}|\mathbf{A}\mathbf{x}_0, \mathbf{\Sigma}_y)$, this approximation yields an analytic solution for the likelihood score:

$$\nabla_{\mathbf{x}_t} \log p(\mathbf{y}|\mathbf{x}_t) \approx \nabla_{\mathbf{x}_t} \mathbb{E}[\mathbf{x}_0|\mathbf{x}_t]^\top \mathbf{A}^\top \left( \mathbf{\Sigma}_y + \mathbf{A} \mathbb{V}[\mathbf{x}_0|\mathbf{x}_t] \mathbf{A}^\top \right)^{-1} \left( \mathbf{y} - \mathbf{A} \mathbb{E}[\mathbf{x}_0|\mathbf{x}_t] \right) . \tag{9}$$

The moment matching posterior sampling (MMPS; [40]) approximation leads to the best sampling when compared on linear inverse problems. The trick behind MMPS is to use Tweedie's variance formula:

$$\mathbb{V}[\mathbf{x}_0|\mathbf{x}_t] = \mathbf{\Sigma}_t \nabla_{\mathbf{x}_t}^\top \mathbb{E}[\mathbf{x}_0|\mathbf{x}_t]. \tag{10}$$

While the Jacobian, $\nabla_{\mathbf{x}_t}^\top \mathbb{E}[\mathbf{x}_0|\mathbf{x}_t]$, in Equation 10 would be extremely costly to materialize, the MMPS method avoids instantiating the matrix by the use of the vector-Jacobian product combined with a conjugate gradient solver [52] for the inverse in Equation 9.

**Expectation Maximization:** Expectation-maximization (EM) is a framework for finding the maximum likelihood estimate for model parameters, $\theta$, in the presence of hidden variables [53]. EM builds a sequence of model parameters $\theta_0, \theta_1, \ldots, \theta_K$ that monotonically improve the likelihood of the data[3]. We adopt the Monte Carlo EM (MCEM) framework from [40]. We embed a diffusion model prior into an MCEM framework where the hidden variables are the true signals, $\mathbf{x}$, and the observations are the noisy, linear transformations of the signal, $\mathbf{y} = \mathbf{A}\mathbf{x} + \eta$. The following two steps of the framework are then repeated until convergence:

- Expectation (E): Given the current diffusion model parameters $\theta_k$, sample from diffusion posterior for each observation: $\mathbf{x}_i \sim q_{\theta_k}(\mathbf{x}_i|\mathbf{y}_i, \mathbf{A}_i)$. Here $q_{\theta_k}(\mathbf{x}_i|\mathbf{y}_i, \mathbf{A}_i)$ is the distribution given by sampling from the reverse SDE in Equation 4 while using the posterior score from Equation 6 and the denoiser model $d_{\theta_k}(\mathbf{x}_t, t)$.
- Maximization (M): Given the set of posterior samples for the full dataset, $\{\mathbf{x}_i\}$, maximize the data likelihood with respect to the model parameters $\theta$ to get $\theta_{k+1}$. In practice, since the mixing matrix and noise covariance are fixed, the denoising score matching objective (Equation 5) can be used as a surrogate.

---

[3]Note that the MCEM framework used in this work does not give this theoretical guarantee.

# 4 Methods

Our goal is to learn the prior distribution $p(\mathbf{x}^\beta)$ of each source $\{\mathbf{x}^\beta\}_{\beta=1}^{N_s}$ given the observations, $\{\mathbf{y}_{i_\alpha}^\alpha\}$, known mixing matrices, $\mathbf{A}_{i_\alpha}^{\alpha\beta}$, and known noise covariance $\mathbf{\Sigma}_{i_\alpha}^\alpha$ (see Section 2). The prior distribution for each source $\beta$ will be parameterized by a variational distribution $q_{\theta^\beta}(\mathbf{x}^\beta)$, defined by a denoiser diffusion model, $d_{\theta^\beta}(\mathbf{x}_t^\beta, t)$ with parameters $\theta^\beta$. We use an EM framework to iteratively maximize the likelihood of the set of diffusion model parameters $\Theta_k = \{\theta_k^\beta\}_{\beta=1}^{N_s}$, where $k$ indexes the EM round. We summarize the full method, DDPRISM, in Algorithm 1 provided in Appendix A. All of the code to reproduce our method and experiments has been made public[4].

**Maximization Step:** For the M step we want to maximize the expected log-likelihood of the full set of diffusion model parameters with respect to the data with $\mathbf{u}_t^\top = \left[(\mathbf{x}_t^1)^\top, (\mathbf{x}_t^2)^\top, \ldots, (\mathbf{x}_t^{N_s})^\top\right]$:

$$\Theta_{k+1} = \arg\max_\Theta \mathbb{E}_{p(\mathbf{y}^\alpha, \{\mathbf{A}^{\alpha\beta}\}, \mathbf{u}_0)} \left[\log q_\Theta(\mathbf{u}_0, \mathbf{y}^\alpha, \{\mathbf{A}^{\alpha\beta}\})\right] \tag{11}$$

$$= \arg\max_\Theta \mathbb{E}_{p(\mathbf{y}^\alpha, \{\mathbf{A}^{\alpha\beta}\})} \mathbb{E}_{q_{\Theta_k}(\mathbf{u}_0|\mathbf{y}^\alpha, \{\mathbf{A}^{\alpha\beta}\})} \left[\log q_\Theta(\mathbf{u}_0, \mathbf{y}^\alpha, \{\mathbf{A}^{\alpha\beta}\})\right] \tag{12}$$

$$= \arg\max_\Theta \mathbb{E}_{p(\mathbf{y}^\alpha, \{\mathbf{A}^{\alpha\beta}\})} \mathbb{E}_{q_{\Theta_k}(\mathbf{u}_0|\mathbf{y}^\alpha, \{\mathbf{A}^{\alpha\beta}\})} \left[\sum_\beta \log q_{\theta^\beta}(\mathbf{x}_0^\beta)\right], \tag{13}$$

where $q_\Theta(\mathbf{u}_0, \mathbf{y}^\alpha, \{\mathbf{A}^{\alpha\beta}\})$ is the full joint distribution of the observation, mixing matrices, and sources under the diffusion model parameters. In getting from Equation 12 to Equation 13 we have dropped all the terms independent of $\Theta$ and taken advantage of the independence of the sources. Since each distribution $q_{\theta^\beta}(\mathbf{x}^\beta)$ is independent of the others, Equation 13 reduces to optimizing each diffusion model separately on the samples from its corresponding source. We can take an expectation value over $p(\mathbf{y}^\alpha, \{\mathbf{A}^{\alpha\beta}\})$ by drawing examples from the dataset, so all that remains is defining how to sample from the joint posterior distribution $q_{\Theta_k}(\{\mathbf{x}^\beta\}|\mathbf{y}^\alpha, \{\mathbf{A}^{\alpha\beta}\})$ given the current set of diffusion model parameters $\Theta_k$.

**Expectation Step:** We want to sample from the joint distribution $q_{\Theta_k}(\{\mathbf{x}^\beta\}|\mathbf{y}^\alpha, \mathbf{A}^{\alpha\beta})$. To do so we need the joint posterior score:

$$\nabla_{\mathbf{u}_t} \log q_{\Theta_k}(\mathbf{u}_t|\mathbf{y}^\alpha, \{\mathbf{A}^{\alpha\beta}\}) = \nabla_{\mathbf{u}_t} \log q_{\Theta_k}(\mathbf{u}_t) + \nabla_{\mathbf{u}_t} \log q_{\Theta_k}(\mathbf{y}^\alpha|\mathbf{u}_t, \{\mathbf{A}^{\alpha\beta}\}). \tag{14}$$

Because our sources are independent, the first term on the right-hand side of Equation 14 simplifies to:

$$\nabla_{\mathbf{u}_t} \log q_{\Theta_k}(\mathbf{u}_t) = \sum_\beta \nabla_{\mathbf{u}_t} \log q_{\theta_k^\beta}(\mathbf{x}_t^\beta), \tag{15}$$

which is simply the sum of the individual diffusion model scores. The remaining likelihood term in Equation 14 is given by:

$$q_{\Theta_k}(\mathbf{y}^\alpha|\mathbf{u}_t, \{\mathbf{A}^{\alpha\beta}\}) = \int \cdots \int p(\mathbf{y}^\alpha|\{x_0^\beta\}, \{\mathbf{A}^{\alpha\beta}\}) \prod_\beta q_{\theta_k}(x_0^\beta|x_t^\beta)\, \mathrm{d}x_0^\beta. \tag{16}$$

To solve this we employ the MMPS approximation [40], wherein each conditional distribution $q_{\theta_k}(x_0^\beta|x_t^\beta)$ is approximated by its first and second moments. Since the conditional distributions are now Gaussian, Equation 16 is simply $N_s$ analytic Gaussian convolutions. The final likelihood score approximation is then:

$$\nabla_{\mathbf{u}_t} \log q_{\Theta_k}(\mathbf{y}^\alpha|\mathbf{u}_t, \{\mathbf{A}^{\alpha\beta}\}) = \begin{bmatrix} \nabla_{\mathbf{x}_t^1} \mathbb{E}[\mathbf{x}_0^1|\mathbf{x}_t^1]^\top (\mathbf{A}^{\alpha 1})^\top \\ \vdots \\ \nabla_{\mathbf{x}_t^{N_s}} \mathbb{E}[\mathbf{x}_0^{N_s}|\mathbf{x}_t^{N_s}]^\top (\mathbf{A}^{\alpha N_s})^\top \end{bmatrix}$$

$$\times \left(\mathbf{\Sigma}_{i_\alpha}^\alpha + \sum_\beta \mathbf{A}^{\alpha\beta} \mathbb{V}[\mathbf{x}_0^\beta|\mathbf{x}_t^\beta](\mathbf{A}^{\alpha\beta})^\top\right)^{-1} \left(\mathbf{y}^\alpha - \sum_\beta \mathbf{A}^{\alpha\beta} \mathbb{E}[\mathbf{x}_0^\beta|\mathbf{x}_t^\beta]\right). \tag{17}$$

---

[4]Code: https://github.com/swagnercarena/DDPRISM

| Method | Posterior | | | | Prior | | |
|---|---|---|---|---|---|---|---|
| | PQM ↑ | FID ↓ | PSNR ↑ | SD ↓ | PQM ↑ | FID ↓ | SD ↓ |
| **1D Manifold: Cont. 2 Sources** | | | | | | | |
| PCPCA [29] | 0.0 | – | 9.35 | 7.69 | 0.0 | – | 7.91 |
| CLVM - Linear [28] | 0.0 | – | 9.58 | 5.80 | 0.0 | – | 5.86 |
| CLVM - VAE [28] | 0.0 | – | 17.15 | 1.81 | 0.0 | – | 2.91 |
| DDPRISM-Gibbs [54] | 0.0 | – | 12.66 | 3.96 | 0.0 | – | 3.92 |
| DDPRISM-Joint [Ours] | **0.26** | – | **38.27** | **0.35** | 0.01 | – | **0.37** |
| **1D Manifold: Cont. 3 Sources** | | | | | | | |
| PCPCA [29] | 0.0 | – | 6.89 | 12.57 | 0.0 | – | 10.22 |
| CLVM - Linear [28] | 0.0 | – | 11.64 | 2.03 | 0.0 | – | 2.16 |
| CLVM - VAE [28] | 0.0 | – | 13.09 | 2.22 | 0.0 | – | 1.82 |
| DDPRISM-Gibbs [54] | 0.0 | – | 9.50 | 4.50 | 0.0 | – | 4.53 |
| DDPRISM-Joint [Ours] | 0.0 | – | **19.78** | **0.75** | 0.0 | – | **0.78** |
| **1D Manifold: Mix. ($f_{mix} = 0.1$)** | | | | | | | |
| DDPRISM-Gibbs [54] | 0.0 | – | 17.69 | 1.84 | 0.0 | – | 1.81 |
| DDPRISM-Joint [Ours] | **0.001** | – | **24.15** | **0.05** | 0.0 | – | **0.04** |
| **GMNIST: Cont. Full-Resolution** | | | | | | | |
| PCPCA [29] | 0.0 | 22.3 | 18.99 | – | 0.0 | 176.0 | – |
| CLVM - Linear [28] | 0.0 | 101.3 | 13.30 | – | 0.0 | 139.9 | – |
| CLVM - VAE [28] | 0.0 | 18.87 | 14.56 | – | 0.0 | 57.67 | – |
| DDPRISM-Joint [Ours] | **1.00** | **1.57** | **25.60** | – | **0.20** | **20.10** | – |
| **GMNIST: Cont. Downsampled** | | | | | | | |
| PCPCA [29] | 0.0 | 121.7 | 14.08 | – | 0.0 | 115.4 | – |
| CLVM - Linear [28] | 0.0 | 199.5 | 12.16 | – | 0.0 | 211.4 | – |
| CLVM - VAE [28] | 0.0 | 1008.0 | 8.48 | – | 0.0 | 737.0 | – |
| DDPRISM-Joint [Ours] | **0.94** | **2.36** | **19.73** | – | **0.06** | **12.63** | – |

Table 1: Comparison of metrics between our methods and baselines for all the experiments shown in the paper. For each metric, the arrow indicates whether larger (↑) or smaller (↓) values are optimal. Our method sets or matches the state-of-the-art for all combinations of experiments and baselines. The posterior metrics are calculated using posterior samples and the true source signals. Since these samples are not independent, it is possible to get large positive PQMass p-values.

Note that the computational cost of Equation 17 scales linearly with the number of sources. As with regular MMPS, the Jacobian can be avoided through the use of the vector-Jacobian product, and the gradient of the variance is ignored. We note that a similar joint posterior equation was concurrently derived by Stevens et al. [25] for removing structured noise using diffusion models.

**Gibbs Sampling:** As an alternative to directly sampling the joint posterior, it is also possible to use a Gibbs sampling method. We derive an extension of the Gibbs diffusion algorithm presented in Heurtel-Depeiges et al. [54] for MVSS in Appendix B. We note that converging to the posterior requires a large number of Gibbs sampling rounds for source distributions with complex structure, thereby rendering the Gibbs sampling approach computationally infeasible for most problems.

**Contrastive MVSS Simplification:** For the contrastive MVSS problem, each new view introduces one new source. In theory, this problem is solved by the generic EM method we have presented. In practice, it can be useful to train the diffusion models sequentially, optimizing $\theta^1$ on observations from view $\alpha = 1$, optimizing $\theta^2$ on observations from view $\alpha = 2$ with $\theta^1$ held fixed, and so on. This limits the computational cost by reducing the number of source models in the joint sampling for all but the final view. However, it discards the information about source $\beta$ present in views $\alpha > \beta$. We summarize this simplified method in Appendix A.

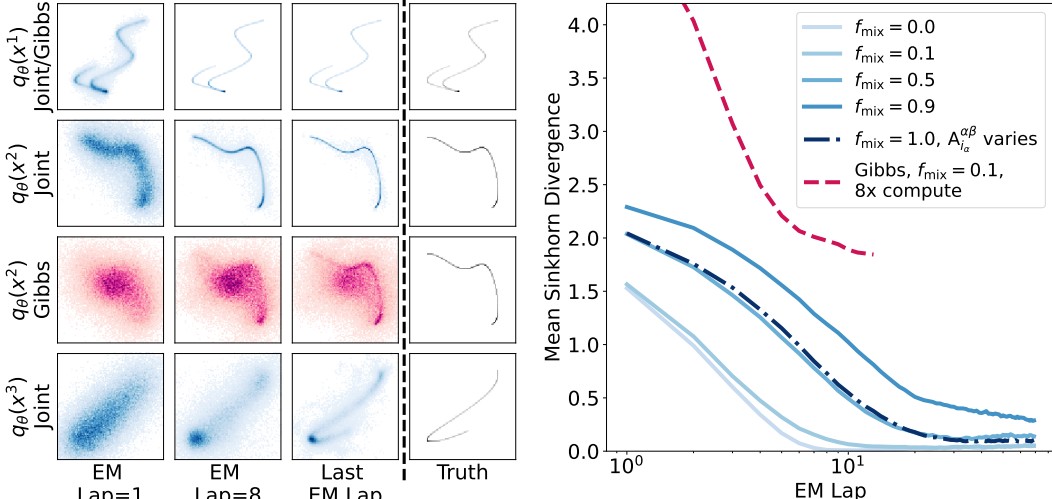

Figure 1: Comparison of posterior samples for our joint sampling method and the Gibbs sampling method [54] on the 1D manifold problem. Both methods are equivalent for the first source. The plots show the evolution of the marginals for the first and second dimension of the specified source distribution. The last EM lap for sources $\beta = 1, 2, 3$ are $16, 32, 64$ respectively.

Figure 2: Comparison of the mean Sinkhorn divergence for different $f_{\text{mix}}$ on the mixed 1D manifold problem. Also shown are the Gibbs sampling method with eight times as many computations per EM lap and our method when $f_{\text{mix}} = 1.0$ and $A_{i_\alpha}^{\alpha\beta}$ depends on $\beta$. Even for large mixing fractions, our method can accurately learn the two distinct underlying source distributions.

## 5 Results

We present five experimental setups that demonstrate the effectiveness of our method. The first four experiments are variations of two synthetic problems previously explored in the literature [26–28, 40]. The final experiment is on the real-world scientific task of separating galaxy light from random light, demonstrating the viability of our method on complex scientific data. Timing comparisons for all experiments can be found in Appendix G.

### 5.1 One-Dimensional Manifold Experiments

In this pair of experiments, our sources, $\mathbf{x}^\beta$, are drawn from distinct, one-dimensional manifolds embedded in $\mathbb{R}^5$. Our observations, $\mathbf{y}^\alpha$, lie in $\mathbb{R}^3$ and are generated by a random linear projection, $\mathbf{A}^{\alpha\beta} \in \mathbb{R}^{3\times5}$, whose rows are drawn from the unit sphere, $\mathbb{S}^4$. In addition, we add isotropic Gaussian noise with standard deviation $\sigma_y = 0.01$ for all views. This setup follows previous work [40], although we now add multiple sources to our observations.

**Contrastive MVSS:** For the contrastive experiment, each view introduces a new source, and the mixing matrix is shared among all the sources: $\mathbf{A}_{i_\alpha}^{\alpha\beta} = \mathbf{A}_{i_\alpha} c^{\alpha\beta}$ with $c^{\alpha\beta} = 1$ if $\beta \leq \alpha$ and $c^{\alpha\beta} = 0$ otherwise. We set $N_{\text{view}} = N_s = 3$ and generate a dataset of size $2^{16}$ for each view.

For our joint diffusion sampling approach, we train three denoiser models, $d_{\theta^\beta}(\mathbf{x}_t^\beta, t)$, each consisting of a multi-layer perceptron. We employ the simplified contrastive MVSS algorithm described in Section 4. We train the $\beta = 1, 2, 3$ diffusion models for $16, 32, 64$ EM laps respectively. For the sampling we use the PC algorithm with 16,384 predictor steps, each with one corrector step (PC step). The initial posterior samples are drawn using a Gaussian prior whose parameters are optimized through a short EM loop. We also train the Gibbs sampling approach with the same denoiser architecture and the same number of EM laps. To keep the computational costs (compute[5]) on par with our joint diffusion approach, we do 64 Gibbs rounds per expectation step and reduce the number of PC steps to 256. Because the Gibbs approach performs poorly, we only run it up to the second view. We also compare to PCPCA, CLVM-Linear, and CLVM-VAE. We provide additional

---

[5]We approximate compute by the number of denoiser and vector-Jacobian product evaluations required.

experimental details on the diffusion parameters, generation of the random manifolds, and baselines in Appendix C.

As shown in Figure 1, our method learns the first two source distributions nearly perfectly despite the linear projection to a lower dimension and the presence of noise. The third source distribution is also learned, although the final posterior samples are not as sharp. This is to be expected: later sources are only observed together with all the previous sources, making them harder to sample. We compare the Sinkhorn divergence [55], PQMass p-value [56], and peak signal-to-noise ratio (PSNR) for our method and the baselines in Table 1. Our method compares favorably, outperforming all the baselines on both source distributions across all the metrics.

**Mixed MVSS:** For the mixed experiment, each source is present in every view. The mixing matrix is given by: $\mathbf{A}_{i_\alpha}^{\alpha\beta} = \mathbf{A}_{i_\alpha} c^{\alpha\beta}$ with $c^{\alpha\beta} = 1$ if $\beta = \alpha$ and $c^{\alpha\beta} = f_{\mathrm{mix}}$ otherwise. We set $N_{\mathrm{views}} = N_s = 2$ and generate a dataset of size $2^{16}$ for each view. We consider four different mixing fractions $f_{\mathrm{mix}} \in \{0.0, 0.1, 0.5, 0.9\}$[6]. When $f_{\mathrm{mix}} = 1.0$ the problem is fully degenerate and therefore not identifiable (see Appendix C). For comparison, we also present a mixed experiment with $\mathbf{A}_{i_\alpha}^{\alpha\beta}$ drawn separately, meaning that every source is fully present in every view but with a different mixing.

For the joint sampling and Gibbs sampling approach, we use the same denoiser models and initialization procedure as the Contrastive MVSS problem. However, because the Gibbs sampling was not able to learn either source distribution with equivalent compute, we instead used 64 Gibbs rounds and 2048 PC steps. This means that each EM lap for the Gibbs sampling is eight times as expensive. We provide additional experimental details in Appendix C.

In Figure 2 we compare the Sinkhorn divergence averaged over both source distributions as a function of EM laps. We find that our method can learn the underlying source distributions with high accuracy up to $f_{\mathrm{mix}} = 0.5$. For $f_{\mathrm{mix}} = 0.9$ our method continues to improve its estimate of the source distributions as the EM laps progress, but it does not converge. If the mixing matrix varies between the sources, we can reconstruct the source distributions even with $f_{\mathrm{mix}} = 1.0$. By comparison, Gibbs sampling for $f_{\mathrm{mix}} = 0.1$ converges much more slowly despite requiring eight times as much compute per EM lap.

## 5.2 Grassy MNIST Experiments

For this pair of experiments, we use the Grassy MNIST dataset first presented in [27]. The dataset is a contrastive MVSS problem which consists of two views: the first containing random $28 \times 28$ crops of grass images from ImageNet [57], and the second containing a linear combination of grass images with 0 and 1 MNIST digits [58]. In addition, we add a small amount of Gaussian noise to each observation ($\sigma_y = 0.01$). For both experiments, we generate 32,768 observations for the grass view and 13,824 for the linear combination of digits and grass.

**Full-Resolution:** In the full-resolution experiment, we set $\mathbf{A}_{i_1}^{11} = \mathbf{A}_{i_2}^{21} = \mathbb{I}$ for $\beta = 1$ (grass) and $\mathbf{A}_{i_1}^{12} = 0$, $\mathbf{A}_{i_2}^{22} = 0.5 \times \mathbb{I}$ for $\beta = 2$ (MNIST). Two example observations can be seen in Figure 3. For our denoiser models, $d_{\theta\beta}(\mathbf{x}_t^\beta, t)$, we use a U-Net architecture [42, 59] with attention blocks [60] and adaLN-Zero norm modulation [61]. We employ the simplified contrastive MVSS algorithm described in Section 4, and we initialize our posterior samples using a Gaussian prior whose parameters are optimized through 32 EM laps. We train the grass and MNIST diffusion models for 64 EM laps. For the sampling we use the PC algorithm with 256 PC steps. We compare to PCPCA, CLVM-Linear, and CLVM-VAE but omit Gibbs sampling since its computational cost makes it impractical for this problem. We provide additional experimental details in Appendix D.

In Table 1 we compare the FID scores [62], the PSNR, and the PQMass p-value on the posterior MNIST digit samples across the entire dataset of the MNIST + grass view. For the FID score, we use a trained MNIST classifier in place of the Inception-v3 network. We also report the FID score and PQMass p-value on samples from the learned priors. In addition, in left-hand side of Figure 3 we show example posterior draws for our method, PCPCA, and CLVM-VAE.

Our method visually returns the closest posterior samples to the ground truth, and outperforms the baselines across all the metrics. The prior samples also outperform the baselines on both PQMass p-value and FID. To better understand the source of this improvement, we run ablation studies over the

---

[6]For $f_{\mathrm{mix}} = 0.0$ we no longer have a source separation problem, but we include this setup as a limiting case.

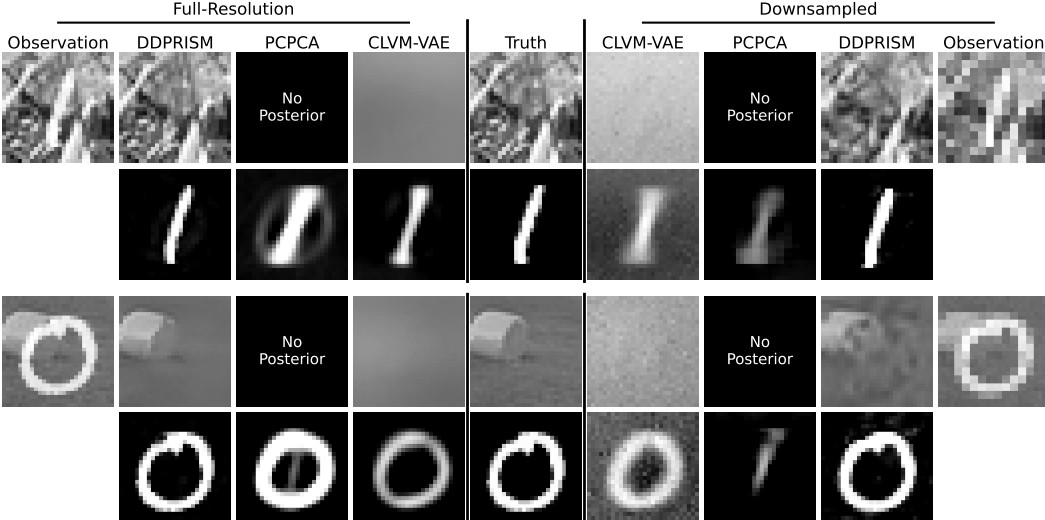

Figure 3: Comparison of posterior samples for two example observations in Grassy MNIST experiment. The observations are on the far left and right, the true input sources are in the middle, and a draw from DDPRISM [ours], CLVM-VAE [28], and PCPCA [29] for both the full-resolution and downsampled case is shown in between. PCPCA cannot sample the grass posterior, and CLVM-Linear is omitted for brevity. Our joint diffusion model returns the best reconstruction of both sources, with near-perfect posterior samples in the full-resolution case.

model architecture, the number of EM laps, the number of initialization laps for the Gaussian prior, the dataset size, and the number of sampling steps. The full ablation study details and results can be found in Appendix H. Overall, the method is fairly insensitive to changes in these hyperparameters. Only extreme choices, such as replacing the U-Net with a small MLP (FID=49.03), conducting only 2 laps of EM (FID=96.85), removing the Gaussian prior initialization entirely (FID=10.41), or using only 16 PC sampling steps (FID=5.41) appear to meaningfully reduce performance across our metrics. The one exception is reductions in the dataset size, where using 1/4th, 1/16th, and 1/64th of the dataset leads to an FID of 4.64, 10.19, and 38.34 respectively.

**Downsampled:** In the downsampling experiment, one third of our observations are at full-resolution, one third are 2x downsampled, and one third are 4x downsampled. Otherwise the dataset size and underlying source distributions are kept the same. An example observation can be seen in Figure 3. We compare to PCPCA, CLVM-Linear, and CLVM-VAE, and use the same configurations as the full-resolution experiment for all four methods. We provide additional details in Appendix D.

In Figure 3 we show example posteriors for 2x downsampling, and in Table 1 we report the FID score, the PSNR, and the PQMass p-value for the posterior samples across the dataset. We also report the FID score and PQMass p-value of draws from the prior distribution. As with the full-resolution dataset, our method visually returns the closest posterior samples to the ground truth for both sources and is SOTA across the baselines. Notably, while both of the CLVM methods and PCPCA struggle with the downsampled images, our method returns visually plausible digits and comparable metric performance to the full-resolution experiment.

## 5.3   Galaxy Images

In astronomical images, instrumental noise, cosmic rays, and random foreground and background objects along the line of sight contaminate observations ("random" light). Separating the flux of these contaminants from the target object is a contrastive MVSS problem. There are two source populations: random light and galaxies. We also get two views: random sightlines that are uncorrelated with galaxies, and targeted sightlines built from a catalog. The targeted sightlines contain both the galaxy and random light. To build our views we use archival Hubble Space Telescope observations of the COSMOS survey [63]. For the galaxy view, we select targets using the Galaxy Zoo Hubble object catalog [64], and for the random view we make cutouts at random locations in

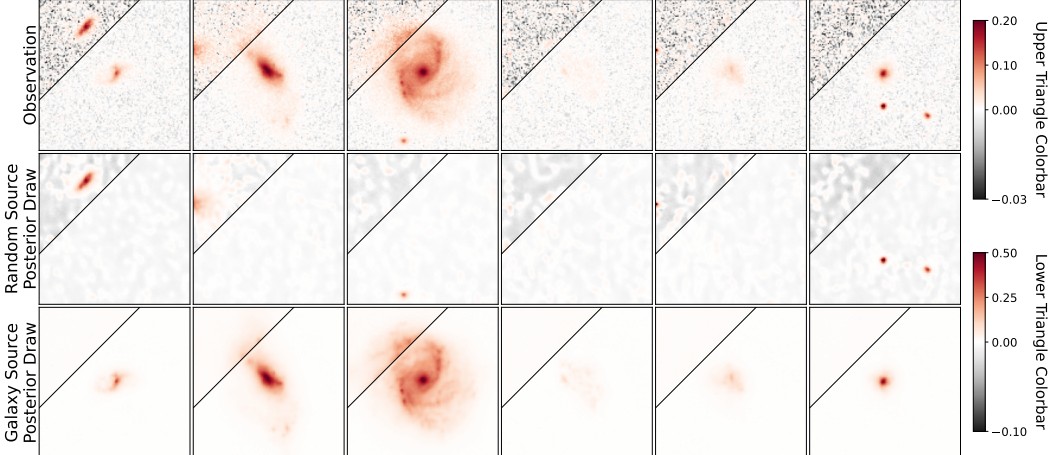

Figure 4: Observations of galaxy images along with posterior samples for the random and galaxy source using our method. Images are split into high-contrast (upper region) and low-contrast (lower region) colormaps to highlight the range of features. The galaxy source captures the central light while the small-scale fluctuations and the uncorrelated light is separated into the random source.

the COSMOS field. Each cutout is $128 \times 128$ pixels. For our denoiser models, $d_{\theta^\beta}(\mathbf{x}_t^\beta, t)$, we use the same U-Net architecture as for Grassy MNIST, but change the model depth and size to model the larger images. We employ the simplified contrastive MVSS algorithm, and we train the random diffusion model for 32 EM laps and the galaxy diffusion model for 16 EM laps. For the sampling we use the PC algorithm with 256 PC steps. We provide additional experimental details in Appendix E.

In Figure 4 we show sample galaxy observations along with a random and galaxy source posterior sample for each observation. While we do not have access to the ground truth, a visual inspection of the source posteriors samples show that our model is effectively identifying the central galaxy light and separating it from the random light. Notably, the background around the galaxy light appears to be nearly flat at zero. We make the full dataset of 79k pristine galaxy images publicly available[7]. Generating the full dataset required 34 hours on four NVIDIA H100 GPUs.

## 6 Discussion and Limitations

We present DDPRISM, a data-driven framework for tackling general MVSS problems using diffusion model priors. To our knowledge, it is the first method to provide a unified solution for linear MVSS problems, achieving state-of-the-art performance across diverse experiments. We further demonstrate that DDPRISM delivers high-quality source separation on a complex real-world astrophysical dataset.

Despite these advances, the framework has important limitations. First, it is restricted to linear source combinations, which excludes nonlinear generative processes like occlusion. The Gaussian noise assumption can be relaxed through the inclusion of an additional "noise" source, but only if an extra view is available. We also assume exact knowledge of the mixing matrix, whereas scientific applications often involve probabilistic rather than deterministic mixing. These assumptions limit our generality and motivate extensions that relax linearity, Gaussianity, and deterministic mixing.

Computationally, our method requires expensive sampling that is compounded by the EM-style training. This places limits on the resolution, dataset size, and number of sources that can be feasibly modeled. Our baselines are far cheaper, albeit at the cost of sample quality. Performance also degrades with smaller datasets, creating a tension between the benefits of large datasets and the computational demands of the method. Replacing our initialization method with random initializations degrades sample quality, suggesting that clever initialization methods may improve convergence and alleviate computational bottlenecks. Nevertheless, DDPRISM establishes diffusion-based MVSS as a promising tool for disentangling structured signals across scientific domains.

---

[7]https://doi.org/10.5281/zenodo.17159988

## Acknowledgments and Disclosure of Funding

We would like to thank Shirley Ho, David Spergel, and Francisco Villaescusa-Navarro for insightful discussions during the development of this project. The computational resources used in this work were provided by the Flatiron Institute, a division of the Simons Foundation. The work of SWC is supported by the Simons Foundation. AA acknowledges financial support from the Flatiron Institute's Predoctoral Program, and thanks the LSST-DA Data Science Fellowship Program, which is funded by LSST-DA, the Brinson Foundation, the WoodNext Foundation, and the Research Corporation for Science Advancement Foundation; her participation in the program has benefited this work. SE acknowledges funding from NSF GRFP-2021313357 and the Stanford Data Science Scholars Program.

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

**Algorithm 1:** MVSS WITH JOINT DIFFUSION

---

**Input:** Dataset $\mathcal{D} = \left\{ \mathbf{y}_{i_\alpha}^\alpha, \{\mathbf{A}_{i_\alpha}^{\alpha\beta}\}_{\beta=1}^{N_s}, \boldsymbol{\Sigma}_{i_\alpha}^\alpha \right\}_{\alpha=1}^{N_{\text{views}}}$, number of sources $N_s$, number of views
$N_{\text{views}}$, initial denoiser parameters $\Theta_0 = \{\theta_0^\beta\}_{\beta=1}^{N_s}$, number of EM rounds $K$

**Output:** Trained diffusion model priors with denoiser parameters $\Theta_K$

**for** $k \leftarrow 0$ *to* $K - 1$ **do**
    **foreach** $\left( \mathbf{y}_{i_\alpha}^\alpha, \{\mathbf{A}_{i_\alpha}^{\alpha\beta}\}, \boldsymbol{\Sigma}_{i_\alpha}^\alpha \right) \in \mathcal{D}$ **do**
        $\left\{ \mathbf{x}_{i_\alpha}^\beta \right\}_{\beta=1}^{N_s} \sim q_{\Theta_k}\left( \{\mathbf{x}^\beta\} | \mathbf{y}_{i_\alpha}^\alpha, \{\mathbf{A}_{i_\alpha}^{\alpha\beta}\} \right)$ `// E step using equation 14`
    **end**
    $\Theta_{k+1} = \arg\max_\Theta \left[ \sum_{i_\alpha} \sum_\beta \log q_{\theta^\beta}(\mathbf{x}_{i_\alpha}^\beta) \right]$ `// M step using equation 5`
**end**
**return** $\Theta_K$

---

---

**Algorithm 2:** SIMPLIFIED CONTRASTIVE MVSS WITH JOINT DIFFUSION

---

**Input:** Dataset $\mathcal{D} = \left\{ \mathbf{y}_{i_\alpha}^\alpha, \{\mathbf{A}_{i_\alpha}^{\alpha\beta}\}_{\beta=1}^{N_s}, \boldsymbol{\Sigma}_{i_\alpha}^\alpha \right\}_{\alpha=1}^{N_{\text{views}}}$, $\mathbf{A}_{i_\alpha}^{\alpha\beta} = 0$ if $\beta > \alpha$, number of sources $N_s$,
number of views $N_{\text{views}} = N_s$, initial denoiser parameters $\Theta_0 = \{\theta_0^\beta\}_{\beta=1}^{N_s}$, number of
EM rounds $K$

**Output:** Trained diffusion model priors with denoiser parameters $\Theta_K$

**for** $a \leftarrow 1$ *to* $N_{views}$ **do**
    **for** $k \leftarrow 0$ *to* $K - 1$ **do**
        **foreach** $\left( \mathbf{y}_{i_\alpha}^\alpha, \{\mathbf{A}_{i_\alpha}^{\alpha\beta}\}, \boldsymbol{\Sigma}_{i_\alpha}^\alpha \right) \in \mathcal{D}$ *with* $\alpha = a$ **do**
            $\left\{ \mathbf{x}_{i_\alpha}^\beta \right\}_{\beta=1}^{a} \sim q_{\{\theta_k^\beta\}_{\beta=1}^{a}}\left( \{\mathbf{x}^\beta\} | \mathbf{y}_{i_\alpha}^\alpha, \{\mathbf{A}_{i_\alpha}^{\alpha\beta}\} \right)$ `// E step using equation 14`
        **end**
        $\theta_{k+1}^a = \arg\max_{\theta^a} \left[ \sum_{i_\alpha} \log q_{\theta^a}(\mathbf{x}_{i_\alpha}^a) \right]$ `// M step for only source a`
    **end**
**end**
**return** $\Theta_K$

---

## A  Joint Sampling Algorithms

In Algorithm 1 we present an algorithmic summary of our MVSS method using joint sampling. This method conducts a joint EM training for all sources simultaneously. In Algorithm 2 we present a summary of the contrastive MVSS simplification. This method trains each source sequentially under the assumption that view $\alpha = 1$ only contains source $\beta = 1$, view $\alpha = 2$ only contains sources $\beta \in \{1, 2\}$, and so on (see Section 4). While Algorithm 2 discards the information about source $\beta$ for views $\alpha \neq \beta$, it reduces the computational complexity.

## B  Gibbs Sampling

Gibbs sampling with diffusion models, originally proposed for blind denoising [54], can be easily extended to the full MVSS problem. We start by selecting a source $\beta_g$. Given the $t = 0$ source realizations for $\{\mathbf{x}_0^{\beta'}\}_{\beta' \neq \beta_g}$, an observation $\mathbf{y}^\alpha$, a set of known mixing matrices $\{\mathbf{A}^{\alpha\beta}\}_{\beta=1}^{N_s}$, and the noise covariance $\boldsymbol{\Sigma}^\alpha$, then the posterior score for the remaining source becomes:

$$
\begin{aligned}
\nabla_{\mathbf{x}_t^{\beta_g}} \log p(\mathbf{x}_t^{\beta_g} | \mathbf{y}^\alpha, \{\mathbf{A}^{\alpha\beta}\}_{\beta=1}^{N_s}, \{\mathbf{x}_0^{\beta'}\}_{\beta' \neq \beta_g}) =& \nabla_{\mathbf{x}_t^{\beta_g}} \log p(\mathbf{x}_t^{\beta_g}) \quad + \\
& \nabla_{\mathbf{x}_t^{\beta_g}} \log p(\mathbf{y}^\alpha | \mathbf{x}_t^{\beta_g}, \{\mathbf{A}^{\alpha\beta}\}_{\beta=1}^{N_s}, \{\mathbf{x}_0^{\beta'}\}_{\beta' \neq \beta_g}).
\end{aligned}
\tag{18}
$$

For conciseness, we can introduce the observation residual $\mathbf{r}^\alpha = \mathbf{y}^\alpha - \sum_{\beta' \neq \beta_g} \mathbf{A}^{\alpha\beta'} \mathbf{x}_0^{\beta'}$. We can then write the remaining likelihood term in Equation 18 as:

$$p(\mathbf{y}^\alpha | \mathbf{x}_t^{\beta_g}, \{\mathbf{A}^{\alpha\beta}\}_{\beta=1}^{N_s}, \{\mathbf{x}^{\beta'}\}_{\beta' \neq \beta_g}) = \int p(\mathbf{r}^\alpha | \mathbf{x}_0^{\beta_g}) p(\mathbf{x}_0^{\beta_g} | \mathbf{x}_t^{\beta_g}) d\mathbf{x}_0^{\beta_g}, \tag{19}$$

which is identical to Equation 7 with the observation replaced by the residual. Since Gibbs sampling fixes all but one source at a time, the posterior evaluation reduces to traditional posterior sampling with a diffusion prior with the observation replaced by the residual. An implementation of the Gibbs sampling procedure for MVSS can be found in the provided code.

## C   One-Dimensional Manifold Experiment Details

We generate random one-dimensional manifolds following the steps outlined in [65]. In all experiments in Section 5.1, the smoothness parameters for the manifolds corresponding to the first, second, and third source distributions are set to $3, 4$, and $5$ respectively. Similarly, we use the same denoiser architecture for each source and each experiment. The denoiser is a multi-layer perceptron (MLP) composed of 3 hidden layers with 256 neurons and SiLU activation function [66], followed by a layer normalization function [67]. The denoiser is conditioned on noise and noise embeddings are generated with the sinusoidal positional encoding method [60].

For the metrics, we use 16384 samples for the Sinkhorn divergence evaluation, 1024 samples for the PQMass evaluation, and 16384 samples for the PSNR evaluation. Note that we use the same number of samples from the true distribution and the prior / posterior distribution for all metrics. Metrics on the posterior samples are calculated by comparing to the true source value for the corresponding observation. For the PQMass evaluation, we use 1000 tessellations and otherwise keep the default parameters.

### C.1   Contrastive MVSS

In the contrastive MVSS experiment, we use the simplified contrastive MVSS algorithm to train the denoiser models. We train the $\beta = 1, 2, 3$ denoiser models for 16, 32, and 64 EM laps respectively. Following Rozet et al. [40], we reinitialize the optimizer and learning rate after each EM lap while keeping the current denoiser parameters. The MLP takes as input a concatenated vector of the diffused sample at time $t$ and the corresponding noise embedding. We summarize other training hyperparameters in Table 2.

For the Gibbs sampling approach we use the same dataset (up to $\beta = 2$), denoiser architectures and training hyperparameters as for the joint posterior sampling approach. The posterior sampling parameters are set to 64 Gibbs rounds with 256 PC steps, and we maintain one corrector step per predictor step. The number of PC steps is lowered so that the number of denoiser and vector-Jacobian product evaluations per EM lap are roughly equivalent for Gibbs and joint sampling.

For both the Gibbs and joint sampling approaches, the denoisers are initialized using samples from a Gaussian prior. Since we are using our contrastive algorithm, only the diffusion model for the current source, $\beta$, is replaced by the Gaussian prior, and the remaining diffusion models for sources $\beta' < \beta$ are kept to the optimum from the previous views (see Algorithm 2). The posterior is then sampled as usual. To optimize the parameters of this Gaussian prior, we use the same EM procedure as for the diffusion model. The final Gaussian prior is used to generate an initial set of posterior samples which are used to train the initial diffusion model, $d_{\theta_0^\beta}(\mathbf{x}_t, t)$.

For the baselines, we use modified implementations built for incomplete data. For PCPCA, we minimize a loss function defined across our two views, $\beta = \text{bkg, targ}$:

$$\begin{aligned}
\mathcal{L}(\mathbf{W}, \mu) = -\frac{1}{2} &\left( \sum_{i=1}^{N_{\text{targ}}} \log \det(\mathbf{C_i}) + \left(\mathbf{y}_i^{\text{targ}} - \mathbf{A}_i^{\text{targ}}\mu\right)^\top \mathbf{C}_i^{-1} \left(\mathbf{y}_i^{\text{targ}} - \mathbf{A}_i^{\text{targ}}\mu\right) \right) \\
&+ \frac{\gamma}{2} \left( \sum_{j=1}^{N_{\text{bkg}}} \log \det(\mathbf{D_j}) + \left(\mathbf{y}_j^{\text{bkg}} - \mathbf{A}_j^{\text{bkg}}\mu\right)^\top \mathbf{D}_j^{-1} \left(\mathbf{y}_j^{\text{bkg}} - \mathbf{A}_j^{\text{bkg}}\mu\right) \right),
\end{aligned} \tag{20}$$

| Parameter | Contrastive MVSS | Mixed MVSS |
|---|---|---|
| **MLP Parameters** | | |
| Activation | SiLU | SiLU |
| Time Embedding Features | 64 | 128 |
| **Training Parameters** | | |
| Optimizer | Adam | Adam |
| Scheduler | Linear | Linear |
| Initial Learning Rate | $10^{-3}$ | $10^{-4}$ |
| Final Learning Rate | $10^{-6}$ | $10^{-5}$ |
| Gradient Norm Clipping | 1.0 | 1.0 |
| Optimization Steps per EM Lap | 65,536 | 65,536 |
| Batch Size | 1024 | 1024 |
| Gaussian Initialization EM Laps | 16 | 8192 |
| **Sampling Parameters** | | |
| Noise Minimum | $10^{-3}$ | $5 \times 10^{-3}$ |
| Noise Maximum | $10^1$ | $1.5 \times 10^1$ |
| Conjugate Gradient Denominator Minimum | 0.0 | $10^{-3}$ |
| Conjugate Gradient Regularization | 0.0 | $10^{-3}$ |
| Predictor-Corrector (PC) Steps | 16,384 | 16,384 |
| Corrections per PC Step | 1 | 1 |
| PC $\tau$ | $10^{-1}$ | $8 \times 10^{-2}$ |

Table 2: Hyperparameters for denoiser training and sampling on the one-dimensional manifold experiments.

where $\mu$ is the learnable mean parameter for the target source. The views, $\mathbf{y}_j^{\text{bkg}}, \mathbf{y}_i^{\text{targ}}$, are the same as those defined in Section 3 but with the mixing matrix $\mathbf{A}_j^{\text{bkg}}$ applied to the sources underlying $\mathbf{y}_j^{\text{bkg}}$ and the mixing matrix $\mathbf{A}_i^{\text{targ}}$ applied to the sources underlying $\mathbf{y}_i^{\text{targ}}$. Here, $\gamma$ is a tunable parameter that controls the relative importance of the variations in the background and target data. The dependence on the weights, $\mathbf{W}$, comes from the two covariance matrices given by:

$$\mathbf{C}_i = \mathbf{A}_i^{\text{targ}}\mathbf{W}\mathbf{W}^\top \left(\mathbf{A}_i^{\text{targ}}\right)^\top + \sigma_i^2 \mathbb{I} \tag{21}$$

$$\mathbf{D}_j = \mathbf{A}_j^{\text{bkg}}\mathbf{W}\mathbf{W}^\top \left(\mathbf{A}_j^{\text{bkg}}\right)^\top + \sigma_j^2 \mathbb{I}, \tag{22}$$

where $\sigma_i$ is the standard deviation of the observation noise. Note that we only optimize the weights and the mean, since the noise is known. We initialize the weights using the empirical covariance derived from source values given by multiplying the observations with the pseudo-inverse of the mixing matrix. We find that this smart initialization considerably improves the performance of PCPCA on incomplete data.

For CLVM-Linear and CLVM-VAE, we explicitly account for the mixing matrix in both the encoding and decoding steps. For CLVM-Linear, the encoded distribution is given by a small modification to the original equations for the latent variable distributions:

$$\mathbf{\Sigma}_j^{\text{bkg}} = \frac{1}{\sigma_j^2}\left(\sigma_j^2\mathbb{I} + \mathbf{S}^\top\mathbf{S}\right) \tag{23}$$

$$\mu_j^{\text{bkg}} = \frac{1}{\sigma_j^2}\mathbf{\Sigma}_j^{\text{bkg}}\mathbf{S}^\top(\mathbf{y}_j^{\text{bkg}} - \mu^{\text{bkg}}) \tag{24}$$

$$\mathbf{\Sigma}_i^{\text{joint}} = \frac{1}{\sigma_i^2}\left(\sigma_i^2\mathbb{I} + \mathbf{M}^\top\mathbf{M}\right) \tag{25}$$

$$\mu_i^{\text{joint}} = \frac{1}{\sigma_i^2}\mathbf{\Sigma}_i^{\text{joint}}\mathbf{M}^\top(\mathbf{y}_i^{\text{joint}} - \mu^{\text{joint}}), \tag{26}$$

where $\mathbf{\Sigma}_j^{\text{bkg}}$ and $\mu_j^{\text{bkg}}$ are the covariance and mean for the latents of $\mathbf{y}_j^{\text{bkg}}$. The covariance and mean $\mathbf{\Sigma}_i^{\text{joint}}$ and $\mu_i^{\text{joint}}$ correspond to the joint latents $(\mathbf{z}^{\text{bkg}}, \mathbf{z}^{\text{targ}})$ for $(\mathbf{y}_i^{\text{joint}})^\top = [(\mathbf{y}_i^{\text{bkg}})^\top, (\mathbf{y}_i^{\text{targ}})^\top]$. The

| Parameter | Grassy MNIST | Galaxy |
|---|---|---|
| **U-Net Parameters** | | |
| Channels per Level | (32, 64, 128) | (64, 128, 256, 256, 512) |
| Residual Blocks per Level | (2, 2, 2) | (2, 2, 2, 2, 2) |
| Attention Heads per Level | (0, 0, 4) | (0, 0, 4, 8, 16) |
| Dropout Rate | 0.1 | 0.1 |
| Activation | SiLU | SiLU |
| **Training Parameters** | | |
| Optimizer | Adam | Adam |
| Scheduler | Cosine Decay | Cosine Decay |
| Initial learning rate | $10^{-3}$ | $10^{-5}$ |
| Gradient Norm Clipping | 1.0 | 1.0 |
| Optimization Steps per EM Lap | 4096 | 4096 |
| Batch size | 1920 | 64 |
| Gaussian Initialization EM laps | 32 | 4 |
| **Sampling Parameters** | | |
| Noise Minimum | $10^{-4}$ | $10^{-2}$ |
| Noise Maximum | $10^2$ | $10^1$ |
| Conjugate Gradient Denominator Minimum | $10^{-2}$ | $10^{-3}$ |
| Conjugate Gradient Regularization | $10^{-6}$ | $10^{-2}$ |
| Conjugate Gradient Error Threshold | $5 \times 10^{-2}$ | $10^1$ |
| Predictor-Corrector (PC) Steps | 256 | 64 |
| Corrections per PC Step | 1 | 1 |
| PC $\tau$ | $10^{-2}$ | $10^{-1}$ |
| EMA decay | 0.999 | 0.995 |

Table 3: Hyperparameters for denoiser training and sampling for the Grassy MNIST experiments and the Galaxy experiment. During sampling, conjugate gradient calculations whose total residuals exceed the conjugate gradient error threshold are recalculated using the denominator minimum and regularization.

matrices $\mathbf{S}, \mathbf{W}$ are the background and target factor loading matrix in CLVM, and $\mathbf{M}$ is the concatenated factor loading matrix, $\mathbf{M}^\top = [\mathbf{S}^\top, \mathbf{W}^\top]$. For CLVM-VAE, the encoded distribution is calculated by explicitly passing in the mixing matrix to the encoder network. For both CLVM-Linear and CLVM-VAE, the decoder outputs the complete source signal and the variational loss is calculated by first transforming the sources using the known mixing matrices. The optimization is done by maximizing the evidence lower bound as described in [28].

For PCPCA, CLVM-Linear, and CLVM-VAE, we optimize the hyperparameters using a 100 point sweep with the default Bayesian optimization used in optuna [68]. The results are reported using the best hyperparameters for each method on the posterior Sinkhorn divergence for three sources. For PCPCA this is $\gamma = 0.3$ and 5 latent dimensions using a linear learning rate from $10^{-3}$ to 0.0 over 10 epochs[8]. For CLVM-Linear, this was a dimensionality of 4 for the background latents and 5 for the source latents, with a batch size of 1024, a cosine learning rate initialized to $10^{-4}$, and 1024 epochs of training. For CLVM-VAE, the encoder architecture were multi-layer perceptrons composed of 3 hidden layers with 256 neurons and a dropout rate of 0.1 followed by a SiLU activation function and a layer normalization function. The decoders follow the same structure. The CLVM-VAE was trained with a batch size of 1024, a cosine learning rate initialized to $5 \times 10^{-4}$, and 1024 epochs of training.

## C.2   Mixed MVSS

In the mixed MVSS experiment, we use Algorithm 1 to train the diffusion models. We train for 70 EM laps and reinitialize the optimizer and learning rate after each EM lap while keeping the current

---

[8]More iterations hampered performance in the hyperparameter sweep.

| Encoder | Decoder | FID |
|---|---|---|
| MLP | MLP | 18.87 |
| U-Net (full-depth) | MLP | 20.14 |
| U-Net (1 hidden channel) | MLP | 45.37 |
| U-Net (full-depth) | U-Net (full-depth) | 388.26 |
| U-Net (1 hidden channel) | U-Net (1 hidden channel) | 390.27 |

Table 4: Evaluation of different CLVM-VAE encoder-decoder architectures for MVSS trained on Grassy MNIST.

denoiser parameters. To condition on the time embedding, the output of each dense layer is passed through a FiLM layer [69]. We summarize other training hyperparameters in Table 2.

For the Gibbs sampling, only the sampling parameters are changed. To improve performance, the number of Gaussian EM laps is reduced to 16, and we use 512 Gibbs rounds with 256 PC steps. As a result, each EM lap of the Gibbs model is eight times as expensive as the joint sampling laps, so we only run 13 laps of EM.

For both the Gibbs and joint sampling approaches, the denoisers are initialized using samples from a Gaussian prior. Unlike for the contrastive algorithm, all of the diffusion models are replaced by the Gaussian prior for initialization. The posterior is then sampled as usual. To optimize the parameters of these Gaussian priors, we use the same EM procedure as for the diffusion model. Note that the resulting Gaussian prior will differ by source. The final Gaussian priors are used to generate an initial set of posterior samples which are used to train the initial diffusion models, $d_{\theta_0^\beta}(\mathbf{x}_t, t)$ for all $\beta$.

### C.2.1 Identifiability

In the case where $f_{\text{frac}} = 1$, the mixed MVSS problem cannot be solved. Consider the vector $\mathbf{z}^1 = \mathbf{x}^1 + \mathbf{x}^2$ and the trivial vector $\mathbf{z}^2 = 0$. Then, for any observation $i_\alpha$, we can write:

$$\mathbf{y}_{i_\alpha}^\alpha = \mathbf{A}_{i_\alpha} \left( \mathbf{x}_{i_\alpha}^1 + \mathbf{x}_{i_\alpha}^2 \right) + \eta_{i_\alpha}^\alpha \tag{27}$$

$$= \mathbf{A}_{i_\alpha} \left( \mathbf{z}_{i_\alpha}^1 + \mathbf{z}_{i_\alpha}^2 \right) + \eta_{i_\alpha}^\alpha. \tag{28}$$

However, $p(\mathbf{z}^2) = \delta(\mathbf{z}^2)$. Since we have constructed $\mathbf{x}^1$ and $\mathbf{x}^2$ to lie on 1D manifolds, we know $p(\mathbf{z}^2) \neq p(\mathbf{x}^1)$ and $p(\mathbf{z}^2) \neq p(\mathbf{x}^2)$. We have found two new sources that match our observations even though one of the sources is guaranteed to not have the same distribution as either of the original sources. Therefore, there is not a unique solution to the source priors when $f_{\text{frac}} = 1$.

## D  Grassy MNIST Experiments

The MNIST dataset is available under a CC BY-SA 3.0 license [58]. The ImageNet dataset is made available for non-commercial purposes [57]. We generate our grass images by taking random $28 \times 28$ pixel crops from ImageNet images with the grass label. We use a different set of random crops for each view. For our digits, we use images with the 0 and 1 label. For both the grass and MNIST images, we normalize the pixel values to the range $[0, 1]$.

We use the same U-Net denoiser architecture for each source and each experiment. The denoiser and training parameters are presented in Table 3. For sampling, we use an exponential moving average of the model weights. The full sampling and U-Net code is provided with out codebase. For the initialization, we use a Gaussian prior optimized via EM as described in Appendix C.

For the metrics, we use 8192 for the FID evaluation, 512 samples for the PQMass evaluation, and 8192 samples for the PSNR evaluation. We use the same number of samples from the true distribution and the prior / posterior distribution for all metrics. Metrics on the posterior samples are calculated by comparing to the true source value for the corresponding observation. For the PQMass evaluation, we use 1000 tessellations and otherwise keep the default parameters.

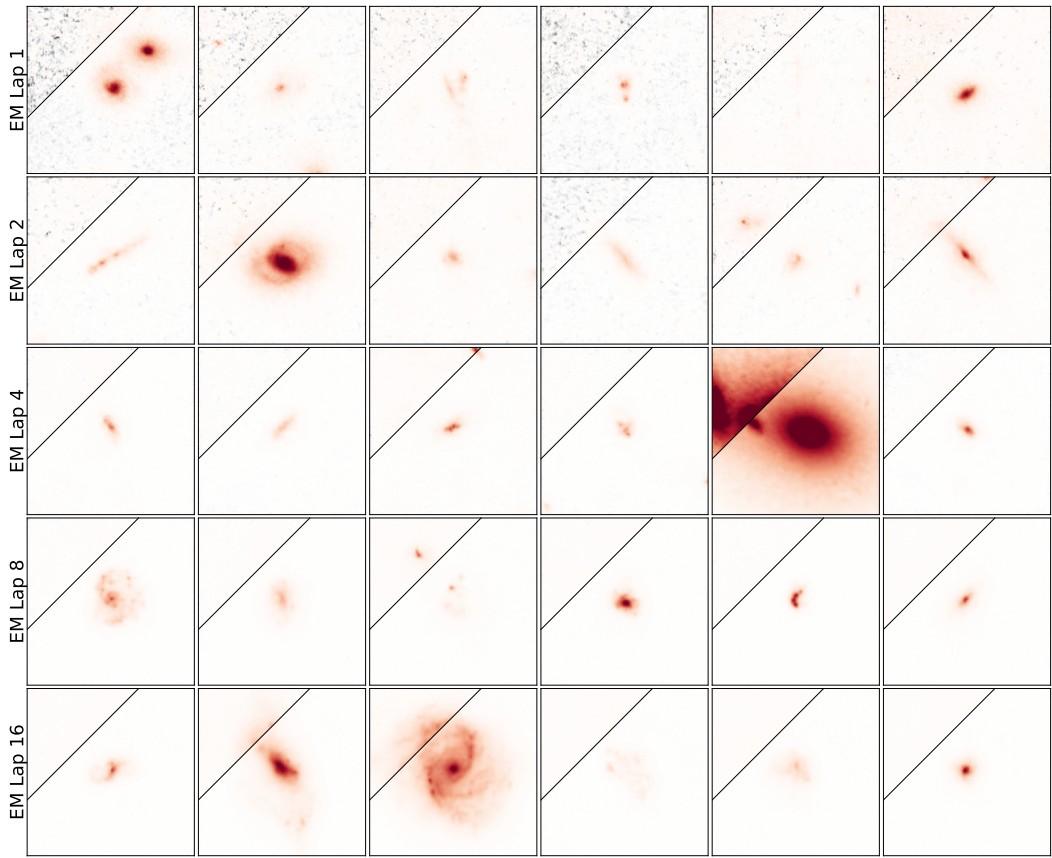

Figure 5: Random samples of galaxy posteriors across EM iterations in the galaxy–image experiment. Early iterations fail to isolate galaxy light and show strong small-scale fluctuations. By iteration 4, small-scale fluctuations are separated but residual uncorrelated light remains. By iteration 16, the uncorrelated light is assigned to the random posterior component, leaving nearly noiseless, isolated galaxy posteriors. As in Figure 4, images are split into high-contrast and low-contrast colormaps to highlight the range of features.

## D.1 Grassy MNIST Baselines

We optimize the PCPCA, CLVM-Linear, and CLVM-VAE hyperparameters using a 100 point sweep with the default Bayesian optimization used in optuna. The results we present are using the best hyperparameters on the FID for the full-resolution experiment.

For PCPCA, we use the traditional algorithm on the full-resolution experiment. On the downsampled experiment we fit the PCPCA parameters on the full-resolution subset and then sample using the equation for incomplete data. The PCPCA tunable parameter is set to $\gamma = 0.39$ and we use 5 latent dimensions.

For CLVM-Linear and CLVM-VAE, we use the traditional algorithm on the full-resolution experiment. On the downsampled experiment we follow the same procedure outlined in Appendix C.1. For CLVM-Linear, we set the dimensionality of the background latents to 265 and the dimensionality of the target latents to 6. We use a cosine learning rate with initial value $1.2 \times 10^{-5}$ trained over batches of 1920 images for $2^{14}$ steps. For CLVM-VAE, we set the dimensionality of the background latents to 380 and the dimensionality of the target latents to 15. The cosine learning rate is used again, but now with an initial value of $2 \times 10^{-4}$, a batch size of 256, and a total of $2^{14}$ steps.

The CLVM-VAE method uses an MLP with three hidden layers of size 70 for the encoder and decoder. The number of hidden layers was optimized during the hyperparameter sweep, but the

| | Training Time | | | Inference Time / Sample | | |
|---|---|---|---|---|---|---|
| Method | 1D Manifold | GMNIST Full Res. | Galaxy Images | 1D Manifold | GMNIST Full Res. | Galaxy Images |
| PCPCA [29] | 5 s | 1 m | – | < 0.1 ms | 1.5 ms | – |
| CLVM - Linear [28] | 1.5 m | 10 m | – | < 0.1 ms | 1 ms | – |
| CLVM - VAE [28] | 18 m | 33 m | – | < 0.1 ms | 1 ms | – |
| DDPRISM-Joint [Ours] | 32 h | 68 h | 48 h | 22 ms | 90 ms | 1.5 s |

Table 5: Comparison of computational costs for our method and baselines for three experiments shown in the paper: contrastive 1D manifold experiment with 2 source distributions, full-resolution Grassy MNIST experiment, and the galaxy experiment. Training time refers to the amount of time each method takes to train its corresponding model. Inference time per sample is the time a method takes to obtain a single posterior sample for an observation, given a trained model. 1D Manifold and Grassy MNIST experiments were run on A100 GPUs, and Galaxy Image experiments were run on H100 GPUs.

encoder and decoder architectures were selected as part of a separate ablation study whose results are summarized in Table 4. In the ablation study, we compared three encoder choices:

- A fully connected MLP.

- A full-depth U-Net identical to the "downsampling" half of U-Net used for our diffusion models without skip connections.

- A convolutional neural network equivalent to our U-Net implementation with 1 hidden channel (no downsampling).

We also compared two decoder choices:

- A fully connected MLP.

- A convolutional neural network equivalent to our U-Net implementation with 1 hidden channel (no upsampling)

We found that more complex architectures negatively impacted performance, and that the best performance was achieved with a simple MLP decoder and encoder.

## E  Galaxy Images Experiment

We query data hosted by the Mikulski Archive for Space Telescopes (MAST), which is available in the public domain. We retrieve data files using the astroquery package, which has a 3-clause BSD style license. We generate our galaxy images by querying $128 \times 128$ pixel cutouts centered on the Galaxy Zoo Hubble Catalog [64]. We generate our random fields by making $128 \times 128$ cutouts at random coordinates within the larger COSMOS exposures. This results in 78,707 galaxy images and 257,219 random images. We apply three normalizations in this order: (1) we pass the images through an arcsinh transform with a scaling of $0.1$, (2) we scale the data by a factor of $0.2$, and (3) we clip the maximum absolute pixel value to 2.0. These three transformations help preserve the morphological features in the brightest sources while stabilizing the diffusion model training.

We use the same U-Net denoiser architecture for the galaxy and random source. The denoiser and training parameters are presented in Table 3. For sampling, we use an exponential moving average of the model weights. For initialization, we use a Gaussian prior optimized via EM as described in Appendix C.1. All of the code required to reproduce this experiment can be found in the DDPRISM codebase. For completeness, in Figure 5 we show the evolution of the galaxy posterior samples as a function of EM laps.

| GMNIST Full Res. | Posterior | | | Prior | |
|---|---|---|---|---|---|
| | PQM ↑ | FID ↓ | PSNR ↑ | PQM ↑ | FID ↓ |
| **Model Architecture** | | | | | |
| MLP | 0.05 | 49.03 | 17.93 | 0. | 215.36 |
| U-Net, small | **1.00** | 2.47 | 25.28 | 0.06 | **15.38** |
| U-Net (*default*) | **1.00** | **1.57** | **25.60** | **0.20** | 20.10 |
| **Training Length (EM laps)** | | | | | |
| 2 | 0. | 96.85 | 16.62 | 0. | 199.70 |
| 8 | **1.00** | **0.04** | **27.15** | 0.14 | **17.35** |
| 32 | **1.00** | 2.26 | 25.66 | 0.08 | 27.96 |
| 64 (*default*) | **1.00** | 1.57 | 25.60 | **0.20** | 20.10 |
| **Initialization Laps** | | | | | |
| 0 (random initialization) | 0.97 | 10.41 | 23.35 | 0.01 | 22.22 |
| 4 | **1.00** | **0.00** | 26.80 | 0.15 | 24.33 |
| 16 | **1.00** | **0.00** | **27.02** | **0.21** | **6.31** |
| 32 (*default*) | **1.00** | 1.57 | 25.60 | 0.20 | 20.10 |
| **Dataset Size** | | | | | |
| Full Dataset (*default*) | **1.00** | **1.57** | **25.60** | **0.20** | 20.10 |
| 1/4th Dataset | 1.00 | 4.64 | 23.67 | 0.14 | 21.36 |
| 1/16th Dataset | 0.99 | 10.19 | 20.97 | 0.07 | **15.16** |
| 1/64th Dataset | 0.0 | 38.34 | 15.34 | 0.0 | 36.55 |
| **Sampling Steps** | | | | | |
| 16 | 0.87 | 5.41 | 21.01 | 0. | 30.20 |
| 64 | **1.00** | 0.88 | 25.02 | **0.24** | **3.58** |
| 256 (*default*) | **1.00** | 1.57 | 25.60 | 0.20 | 20.10 |
| 1024 | **1.00** | **0.59** | **26.50** | 0.08 | 7.18 |

Table 6: Ablation study of individual components and parameters of our method on the full-resolution grassy MNIST experiment. For each metric, the arrow indicates whether larger (↑) or smaller (↓) values are optimal. The values used for the Grassy MNIST (GMNIST) experiment in Section 5.2 are denoted as *default* values. While most of the default values correspond to the optimal performance, we find that decreasing the length of the training and using fewer rounds of Gaussian EM improve the overall performance of the method.

# F    Additional Diffusion Model Details

For our diffusion models, we use the preconditioning strategy from Karras et al. [45]. Our variance exploding noise schedule is given by:

$$\sigma(t) = \exp[\log(\sigma_{\min}) + (\log(\sigma_{\max}) - \log(\sigma_{\min})) * t], \tag{29}$$

with minimum noise $\sigma_{\min}$ and maximum noise $\sigma_{\max}$. During training, we sample the time parameter $t$ from a beta distribution with parameters $\alpha = 3$, $\beta = 3$.

# G    Runtime and Computational Cost

We provide a detailed comparison of the computational costs between our method and baselines in Table 5. All timing was done on NVIDIA A100 GPUs with the exception of the galaxy images experiment that was run on H100s. Both 1D manifold experiments used one A100 (40GB) GPU, both Grassy MNIST experiments used four A100 (40GB) GPUs, and the Galaxy Image experiments used four H100 (80GB) GPUs. Our method is more computationally expensive than the baselines, almost entirely due to the cost of sampling from the diffusion model. However, this computational cost comes with significant improvements in performance and sample quality.

# H  Ablation Studies

In order to clarify the importance of the individual components of our method, we conduct a series of ablation studies which are summarized in Table 6. We explored the effect of varying the following components on the performance of the method:

- **Diffusion model architecture:** We find the model architecture to be important for performance: using an MLP-based architecture (5 hidden layers with 2048 hidden features per layer) gives poor results across all metrics. However, scaling down the UNet model (channels per level: (32,64,128)→(16,32), residual blocks per level: (2,2,2)→(2,2), embedding features: 64→16, attention head moved up one level) does not degrade performance appreciably.

- **Training length:** We observe that the model achieves good performance after as few as 8 EM iterations and that longer training leads to a slight degradation in performance.

- **Initialization strategy:** The number of Gaussian EM laps used to generate the initial samples (and train the initial diffusion model) has minimal impact on performance. Only starting from a randomly initialized model considerably reduces the performance of the method.

- **Training dataset size:** We train our method using 1/4th, 1/16th, and 1/64th of the original grass and grass+MNIST datasets. The 1/16th and 1/64th runs are given 1/4th as many EM laps as the original training to account for the smaller dataset, but all other hyperparameters are unchanged. There is a clear degradation in performance as the grass and MNIST datasets are reduced in size. However, even with 1/16th of the original dataset (2048 grass images, 512 MNIST digits) our method still outperforms the baselines run on the full dataset. We note that there exists a few strategies for improving diffusion model performance in the low-data regime [45, 70–73].

- **Sampling steps:** Increasing the number of predictor steps used to sample from the posterior generally improves the performance of our method. However, we find that the method performs well even with the number of sampling steps reduced to 64.

The overall robustness of our method to most ablation highlights that its effectiveness is driven by the ability to directly sample from the posterior given our current diffusion model prior. Because the likelihood is often constraining, this enables us to return high-quality posterior samples even when the prior specified by our diffusion model is not an optimal fit to the empirical distribution. As evidence for this point, we note the large gap in performance between posterior and prior samples on the Grassy MNIST experiments (see Table 1).

