# OpenReview forum: "A Data-Driven Prism: Multi-View Source Separation with Diffusion Model Priors"
_NeurIPS.cc/2025/Conference — NeurIPS 2025 poster_

### Official Review · Reviewer_FLez · 2025-06-23

**Clarity:** 2
**Significance:** 3
**Originality:** 3
**Rating:** 5
**Confidence:** 3

**Summary:**

This paper proposes a novel diffusion-based multi-view source separation algorithm. The authors introduce an EM algorithm that alternates between estimating the separation results and updating the diffusion model parameters, gradually improving both the separation quality and the accuracy of the prior modeling. In the E-step, a diffusion-based posterior sampling method is employed to infer the separated sources from the observations. In the M-step, the separated results are used as an estimate of the clean samples to update the diffusion parameters. The effectiveness of the proposed method is validated on both synthetic and real-world datasets.

**Questions:**

While I am not an expert in this area, I'm curious about the data requirements for training diffusion models. It is well known that diffusion models typically require large-scale training data. In practical multi-view source separation (MVSS) tasks, is it realistic to expect such a volume of data to be available? The reviewer notes that in this paper, most synthetic datasets reach the scale of 10k samples, even though the data is relatively low-dimensional (e.g., 5D vectors or 28×28 images). Can this scale of data be available in real-world MVSS scenarios?

**Ethical Concerns:**

["NO or VERY MINOR ethics concerns only"]

**Final Justification:**

I believe the authors have clearly addressed the concerns regarding the Diffusion Model initialization and dataset size, particularly clarifying that the size of real-world datasets is sufficient to support effective training of Diffusion Models, which ensures the feasibility of the proposed method. I have also read the other reviewers' comments and the authors’ responses, and I think the additional experiments and discussions have substantially improved the quality of the paper. I raise my score to 5, and I believe the methods and findings in this paper could be useful to the NeurIPS audience.

**Limitations:**

Yes

**Quality:**

3

**Strengths And Weaknesses:**

### Strengths:

- The paper is clearly written and easy to follow.
- The simulation experiments suggest that the proposed algorithm consistently outperforms the baselines across all scenarios. (I would like to hedge this claim as I am not entirely familiar with the domain.)
- The problem tackled in this paper is meaningful, and the proposed solution is well-motivated and clearly structured.

### Weaknesses:

- One concern from the reviewer is about the initialization of the diffusion model. According to the provided pseudocode and description, the algorithm first performs posterior sampling using the diffusion model to obtain an initial separation result, which is then used to update the diffusion model in subsequent EM steps. However, if the initial diffusion model is randomly initialized and does not model the prior well, how can the first posterior sampling step be reliable? In other words, if the diffusion model starts from random weights, I believe the posterior sampling algorithm would generate meaningless noise, making it difficult to proceed with the following M and E steps. The reviewer would appreciate it if the authors could clarify how the diffusion model is initialized or pre-trained, or point out if there is a misunderstanding about the algorithmic pipeline. This is important for assessing the feasibility and strengths of the proposed method.
- The reviewer suggests adding an ablation study beyond the toy example to evaluate how increasing or decreasing the number of EM iterations affects modeling performance. This could further enhance the understanding of the proposed approach.
- Minor error: Line 217: it should be $\mathbb{R}^{3\times 5}$ instead of $\mathbb{R}^{3x5}$.

The reviewer looks forward to further discussion.

---

> ### Author Rebuttal · Authors · 2025-07-31
>
> We would like to thank the reviewer for their detailed comments and feedback. Below we address the concerns and questions brought up within the review:
> ### Initialization
>
> As the reviewer points out, the results of EM can depend heavily on the quality of the initialization. EM reaches a local but not necessarily global optimum. To mitigate this issue, we follow [1] and start by running an EM procedure that optimizes the parameters of a simple Gaussian diffusion model (mean and covariance). These approximate samples are then used to train our initial full diffusion model, which we then plug into our EM loop. We agree that exploring the importance of this initialization procedure helps contextualize our results, so we ran an ablation study on the Grassy MNIST experiment where we test random initialization compared to initialization using a Gaussian diffusion model optimized over 4, 16, and 64 (the default) EM laps. Below we show the results of our ablation study:
>
> |||Posterior|||||Prior||
> |-|:-:|:-:|:-:|:-:|:-:|:-:|:-:|:-:|
> |**GMNIST Full Res.**|**PQM ↑**|**FID ↓**|**PSNR ↑**|**SD ↓**||**PQM ↑**|**FID ↓**|**SD ↓**|
> |**Initialization (# of Gaussian EM laps)** |||||||||
> |0 (Random)|0.97|10.41|23.35|--|¦|0.01|22.22|--|
> |4|**1.00**|**0.00**|26.80|--|¦|0.15|24.33|--|
> |16|**1.00**|**0.00**|**27.02**|--|¦|**0.21**|**6.31**|--|
> |64 (Default)|**1.00**|1.57|25.60|--|¦|0.20|20.10|--|
>
> We find that the number of Gaussian EM laps used to generate the initial samples (and therefore train the initial diffusion model) has minimal impact on performance. Only starting from a randomly initialized model considerably reduces performance. While this result may seem surprising, we believe it can be explained by our ability to directly sample from the posterior given our current diffusion model prior (Eq.17). Because the likelihood is often constraining, this enables us to return high-quality posterior samples even when the prior specified by our diffusion model is not an optimal fit to the empirical distribution. As evidence for this point, we note the large gap in performance between posterior and prior samples on the Grassy MNIST experiment. This means that even if the initial prior fit is poor, the resulting posterior samples will likely not be random noise. We included this ablation study in the appendix of our work and pointed to it in Sections 5.2 and 6. We also reworked the introduction of our method in Section 4 and the pseudocode to highlight the use of the Gaussian denoiser initialization.
>
> ### EM Laps
>
> We agree that a full exploration of the performance of our model as a function of EM laps is important, especially given the higher computational cost of our model compared to the baseline methods. Beyond the performance on the one-dimensional manifold problem shown in Figures 1 and 2, we also added a comparison of our model’s performance on the Grassy MNIST experiment as a function of EM laps to the appendix. We summarize the results below:
>
> |||Posterior|||||Prior||
> |-|:-:|:-:|:-:|:-:|:-:|:-:|:-:|:-:|
> |**GMNIST Full Res.**|**PQM ↑**|**FID ↓**|**PSNR ↑**|**SD ↓**||**PQM ↑**|**FID ↓**|**SD ↓**|
> |**Training Length (EM laps)**|||||||||
> |2|0.|96.85|16.62|--|¦|0.|199.70|--|
> |8|**1.00**|**0.04**|**27.15**|--|¦|0.14|**17.35**|--|
> |32|1.00|2.26|25.66|--|¦|0.08|27.96|--|
> |64 (Default)|1.00|1.57|25.60|--|¦|**0.20**|20.10|--|
>
> We find that the model achieves good performance after as few as 8 EM iterations and that longer training leads to a slight degradation in performance. In the table comparing performance to baselines we report the results for the model trained for the default number (64) of EM laps.
>
> For completeness, we also added a Figure to the appendix showing how the visual performance of the galaxy experiment varies as a function of EM Laps. However, given the restrictions of the rebuttal, we cannot reproduce that here.
>
> ### Dataset Size
>
> The reviewer brings up a valid concern about the scale of the dataset required to successfully fit a diffusion model to a distribution. Some previously explored solutions to reduce the dataset size requirement include data augmentation [2], patch-wise diffusion training [3], latent-space diffusion models [4,5], and transfer learning [6]. It would be possible to incorporate these methods during our maximization step.
>
> To better quantify our dependence on the dataset size, we ran a set of ablation studies on our performance on the full-resolution Grassy MNIST experiment. We train our method using 1/4th, 1/16th, and 1/64th of the original grass and grass+MNIST datasets. The 1/16th and 1/64th runs are given 1/4th as many EM Laps as the original training setup to account for the smaller dataset, but all other hyperparameters are unchanged:
>
> |||Posterior|||||Prior||
> |-|:-:|:-:|:-:|:-:|:-:|:-:|:-:|:-:|
> |**GMNIST Full Res.**|**PQM ↑**|**FID ↓**|**PSNR ↑**|**SD ↓**| |**PQM ↑**|**FID ↓**|**SD ↓**|
> |Full Dataset|**1.00**|**1.57**|**25.60**|--|¦|**0.20**|20.10|--|
> |1/4th Dataset|1.00|4.64|23.67| -- |¦|0.14|21.36|--|
> |1/16th Dataset|0.99|10.19|20.97|--|¦|0.07|**15.16**|--|
> |1/64th Dataset|0.0|38.34|15.34|--|¦|0.0|36.55|--|
>
> There is a clear degradation in performance as the grass and MNIST datasets are reduced in size. However, even with 1/16th of the original dataset (2048 grass images, 512 MNIST digits) our method still outperforms the baselines run on the full dataset. We added the dataset scaling study to the appendix and referenced it in the Discussion and Limitations section.
>
> In terms of whether these dataset sizes are realistic for real world applications, we think the galaxy experiment serves as a relevant example. Using just the HST COSMOS field, we have millions of random cutouts and ~80k galaxy image examples. If we incorporate ground-based telescopes or future space-based telescopes, we can access orders-of-magnitude more random and galaxy datapoints [7].
>
> ### Additional notes
>
> * To ensure fair comparison, we ran a hyperparameter sweep for the baseline methods. Similarly, we made small architectural changes to improve the diffusion models used in our method and reran all experiments. Therefore, some of the metrics quoted in the tables above changed from our original submission.
>
> [1] Rozet, François, et al. "Learning diffusion priors from observations by expectation maximization" (2024)
>
> [2] Karras, Tero, et al. "Elucidating the design space of diffusion-based generative models" (2022)
>
> [3] Wang, Zhendong, et al. "Patch diffusion: Faster and more data-efficient training of diffusion models" (2023)
>
> [4] Zhang, Zhaoyu, et al. “Training Diffusion-Based Generative Models with Limited Data” (2025)
>
> [5] Rombach, Robin, et al. "High-resolution image synthesis with latent diffusion models" (2022)
>
> [6] Hur, Jiwan, et al. "Expanding expressiveness of diffusion models with limited data via self-distillation based fine-tuning" (2024)
>
> [7] Mellier, Y., et al. "Euclid. I. Overview of the Euclid mission" (2024)

---

> > ### Comment · Reviewer_FLez · 2025-08-01
> > **Thank you for your rebuttal**
> >
> > Thank you to the authors for the time and effort spent on the rebuttal. I believe the authors have clearly addressed the concerns regarding the Diffusion Model initialization and dataset size, particularly clarifying that the size of real-world datasets is sufficient to support effective training of Diffusion Models, which ensures the feasibility of the proposed method. I have also read the other reviewers' comments and the authors’ responses, and I think the additional experiments and discussions have substantially improved the quality of the paper. I am happy to raise my score to 5.
> >
> > By the way, I found it somewhat surprising that using Gaussian EM for initializing the Diffusion Model could work well, as the Gaussian distribution does not provide any meaningful prior in this context. I believe this aspect deserves further exploration, both in terms of theoretical understanding and in the development of potentially better initialization strategies.

---

### Official Review · Reviewer_G8fL · 2025-07-01

**Clarity:** 3
**Significance:** 3
**Originality:** 3
**Rating:** 5
**Confidence:** 4

**Summary:**

The paper introduces a multi-view source separation (MVSS) method using diffusion model priors. The method learns on data where observations consist of different linear transformations of the unknown sources. The approach works by training separate diffusion models for each source, facilitating sampling and probability estimation. The method is able to handle noisy, incomplete, and observations with varied resolution without requiring explicit source assumptions. It employs an expectation-maximisation framework with moment matching posterior sampling (MMPS) to learn source priors and outperforms existing methods like PCPCA and CVAE in synthetic and real-world datasets. Experimental results demonstrate state-of-the-art performance on one-dimensional manifold and Grassy MNIST datasets, as well as real-world galaxy image separation from the COSMOS survey.

**Questions:**

- Why does their diffusion-based approach work so effectively for separating linear mixtures of sources? This is not fully addressed, e.g. in terms of specific mechanisms related to diffusion models or architectural inductive biases. While the role of diffusion models as expressive priors and the importance of the multi-view setup is emphasised, but the work lacks a detailed explanation of how the diffusion process or the architectural choices aid the separation of sources. A more explicit discussion of the interaction between the linear mixing model, the diffusion process, and the architectural biases would strengthen the response to this question. Furthermore ablations could reinforce any conclusions or speculations on this.
- Could the authors add additional results from other real-world experiments (beyond galaxy separation)? This could help demonstrate the general utility and applicability of the method and broaden impact. E.g. perhaps showing demonstrating results on cell separation in neuroscience or sound source separation etc.
 - Could the work be applied in situations where the mixing matrix is unknown? And have the authors considered whether the method could be meaningfully extended to datasets with non-linear mixtures of sources? Could the method be meaningfully applied in such scenarios, or are there constraints in the assumptions made that preclude this? Such extensions could significantly broaden the applicability of the approach across diverse scientific and real-world domains.

**Ethical Concerns:**

["NO or VERY MINOR ethics concerns only"]

**Final Justification:**

The authors addressed most of my and other reviewer concerns raised. Importantly the authors provided additional baselines and evaluation metrics, ablations results, computational cost analyses, and also shared additional discussion on the effectiveness and limitations of their approach. However, the authors were unable to provide results on other real-world datasets beyond galaxy observations due to time constraints. While adding such results would further extend the scope and impact of this interesting method, the work is now solid enough rate it Accept.

**Limitations:**

Yes

**Quality:**

2

**Strengths And Weaknesses:**

The paper introduces a novel method that uses score-based diffusion models for multi-view source separation, bypassing the need for explicit source assumptions. The approach is innovative as it allows for flexible modelling of complex source distributions, with the potential to disentangle signals in diverse scientific applications. And it leverages the capability of diffusion models to sample complex joint posteriors.

The paper is well-written, with a clear structure, and presents technical concepts and experimental results in a coherent manner. This work has the potential to advance transform source separation tasks across scientific domains, such as astronomy, neuroscience, and seismology, where disentangling mixed signals is critical for scientific progress. The paper demonstrates its method’s effectiveness primarily through a limited set of experiments, including synthetic 1D manifold and Grassy MNIST datasets and one real-world galaxy image separation task. This limits the diversity of tested scenarios and may not fully showcase the method’s generalisability across varied real-world applications. Additionally, the absence of explicit ablation studies, such as testing variations in the diffusion model architecture, sampling strategy, or EM framework components, limits insights into the specific contributions of each element, weakening the understanding of why the method excels for linear mixtures.

While effective, their method is computationally intensive and assumes linear observation models, limiting its applicability to datasets with non-linear characteristics. The authors discuss and address the computational limitations, including heavy compute requirements and being data hungry. They also provide guidance on where existing techniques would be better suited. But the main paper does not provide a straightforward quantification of what the extra burden is relative to other methods.

---

> ### Author Rebuttal · Authors · 2025-07-31
>
> We would like to thank the reviewer for their detailed comments and feedback. Below we address the concerns and questions brought up within the review:
> ### Additional Results
> Running on a fourth dataset was not feasible for the rebuttal, but we ran a wider set of baselines and evaluation metrics to better compare the models performance:
> |||Posterior|||||Prior||
> |-|:-:|:-:|:-:|:-:|:-:|:-:|:-:|:-:|
> |**Dataset / Method**|**PQM ↑**|**FID ↓**|**PSNR ↑**|**SD ↓**||**PQM ↑**|**FID ↓**|**SD ↓**|
> |**1D Manifold: Cont. 2 Sources**|||||||||
> |PCPCA [1]|0.0|--|9.35|7.69|¦|0.0|--|7.91|
> |CLVM - Linear [2]|0.0|--|9.58|5.80|¦|0.0|--|5.86|
> |CLVM - VAE [2]|0.0|--|17.15|1.81|¦|0.0|--|2.91|
> |DDPRISM-Gibbs [3]|0.0|--|12.66|3.96|¦|0.0|--|3.92|
> |DDPRISM-Joint [Ours]|**0.26**|--|**38.27**|**0.35**|¦|**0.01**|--|**0.37**|
> |**1D Manifold: Cont. 3 Sources**||||||||||
> |PCPCA [1]|0.0|--|6.89|12.57|¦|0.0|--|10.22|
> |CLVM - Linear [2]|0.0|--|11.64|2.03|¦|0.0|--|2.16|
> |CLVM - VAE [2]|0.0|--|13.09|2.22|¦|0.0|--|1.82|
> |DDPRISM-Gibbs [3]|0.0|--|9.50|4.50|¦|0.0|--|4.53|
> |DDPRISM-Joint [Ours]|0.0|--|**19.78**|**0.75**|¦|0.0|--|**0.78**|
> |**1D Manifold: Mix. ($f_{mix}=0.1$)**|||||||||
> |DDPRISM-Gibbs [3]|0.0|--|15.48|2.52|¦|0.0|--|2.43|
> |DDPRISM-Joint [Ours]|**0.001**|--|**24.15**|**0.05**|¦|0.0|--|**0.04**|
> |**GMNIST: Cont. Full Resolution**|||||||||
> |PCPCA [1]|0.0|22.30|18.99|--|¦|0.0|176.0|--|
> |CLVM - Linear [2]|0.0|101.3|13.30|--|¦|0.0|139.9|--|
> |CLVM - VAE [2]|0.0|18.87|14.56|--|¦|0.0|57.67|--|
> |DDPRISM-Joint [Ours]|**1.00**|**1.57**|**25.60**|--|¦|**0.20**|**20.10**|--|
> |**GMNIST: Cont. Downsampled**|||||||||
> |PCPCA [1]|0.0|121.7|14.08|--|¦|0.0|115.4|--|
> |CLVM - Linear [2]|0.0|199.5|12.16|--|¦|0.0|211.4|--|
> |CLVM - VAE [2]|0.0|1008.0|8.48|--|¦|0.0|737.0|--|
> |DDPRISM-Joint [Ours]|0.0|**60.34**|**17.09**|--|¦|0.0|**79.43**|--|
>
> PSNR is the peak signal-to-noise ratio, PQM is the pqmass metric introduced in [4], FID is the FID score [5] using our MNIST classifier, and SD is the sinkhorn divergence. Note that we’ve replaced the CVAE [6] baseline with CLVM-VAE [2] since the latter is a superset of the former and achieves better performance. Across the experiments, we find that our method substantially outperforms the baselines.
>
> ### Runtime and Computational Cost
>
> In addition, we provide a detailed comparison of the computational time for each method in the table below:
>
> |||Training Time||¦||Inference Time per Sample||
> |-|:-:|:-:|:-:|:-:|:-:|:-:|:-:|
> |**Method**|**1D Manifold**|**GMNIST Full Res.**|**Galaxy Images**|¦|**1D Manifold**|**GMNIST Full Res.**|**Galaxy Images**|
> |PCPCA [1]|5s|1m|--|¦|<0.1ms|1.5ms|--|
> |CLVM - Linear [2]|1.5m|10m|--|¦|<0.1ms|1ms|--|
> |CLVM - VAE [2]|18m|33m|--|¦|<0.1ms|1ms|--|
> |DDPRISM-Joint [Ours]|32h|68h|48h|¦| 22ms| 90ms|1.5s|
>
> All timing was done on NVIDIA A100 GPUs. Our method is more computationally expensive than the baselines, almost entirely due to the cost of sampling from the diffusion model. However, this computational cost comes with significant improvements in performance and sample quality. In our ablation studies below we also compare the performance as a function of EM laps, which can be used to control the tradeoff between training time and sample quality for our method.
> ### Effectiveness on Linear Source Separation
> The effectiveness of our method is driven by the ability to directly sample from the posterior given our current diffusion model prior (Eq.17). Because the likelihood is often constraining, this enables us to return high-quality posterior samples even when the prior specified by our diffusion model is not an optimal fit to the empirical distribution. As evidence for this point, we note the large gap in performance between posterior and prior samples on the Grassy MNIST experiments.
> ### Ablation Studies
> Per reviewer’s suggestion, in order to clarify the importance of individual components of our proposed method, we conducted an ablation study. In particular, we explored the effect of the following components on the performance of our method: (1) diffusion model architecture; (2) training length (in terms of the number of EM iterations), (3) initialization strategy, and (4) dataset size:
> |||Posterior|||||Prior||
> |-|:-:|:-:|:-:|:-:|:-:|:-:|:-:|:-:|
> |**GMNIST Full Res.**|**PQM ↑**|**FID ↓**|**PSNR ↑**|**SD ↓**||**PQM ↑**|**FID ↓**|**SD ↓**|
> |**Model Architecture**|||||||||
> |MLP|0.05|49.03|17.93|--|¦|0.|215.36|--|
> |UNet, small|1.00|2.47|25.28|--|¦|0.06|**15.38**|--|
> |UNet, default|**1.00**|**1.57**|**25.60**|--|¦|**0.20**|20.10|--|
> |**Training Length (EM laps)**|||||||||
> |2|0.|96.85|16.62|--|¦|0.|199.70|--|
> |8|**1.00**|**0.04**|**27.15**|--|¦|0.14|**17.35**|--|
> |32|1.00|2.26|25.66|--|¦|0.08|27.96|--|
> |64 (Default)|1.00|1.57|25.60|--|¦|**0.20**|20.10|--|
> |**Initialization (# of Gaussian EM laps)** |||||||||
> |0 (Random)|0.97|10.41|23.35|--|¦|0.01|22.22|--|
> |4|**1.00**|**0.00**|26.80|--|¦|0.15|24.33|--|
> |16|**1.00**|**0.00**|**27.02**|--|¦|**0.21**|**6.31**|--|
> |64 (Default)|**1.00**|1.57|25.60|--|¦|0.20|20.10|--|
> |**Dataset Size**|||||||||
> |Full Dataset|**1.00**|**1.57**|**25.60**|--|¦|**0.20**|20.10|--|
> |1/4th Dataset|1.00|4.64|23.67| -- |¦|0.14|21.36|--|
> |1/16th Dataset|0.99|10.19|20.97|--|¦|0.07|**15.16**|--|
> |1/64th Dataset|0.0|38.34|15.34|--|¦|0.0|36.55|--|
>
> We find that the model achieves good performance after as few as 8 EM iterations and that longer training leads to a slight degradation in performance. In the table comparing performance to baselines we report the results for the model trained for the default number of EM laps. For dataset size, we train our method using 1/4th, 1/16th, and 1/64th of the original grass and grass+MNIST datasets. The 1/16th and 1/64th runs are given 1/4th as many EM Laps as the original training to account for the smaller dataset, but all other hyperparameters are unchanged. There is clear degradation in performance as the grass and MNIST datasets are reduced in size. However, even with 1/16th of the original dataset (2048 grass images, 512 MNIST digits) our method still outperforms the baselines run on the full dataset. Surprisingly, the number of Gaussian EM laps used to generate the initial samples (and therefore train the initial diffusion model) has minimal impact on performance. Only starting from a randomly initialized model considerably reduces performance.
>
> We also find that model architecture has significant impact on performance: using an MLP-based architecture (5 hidden layers with 2048 hidden features per layer) gives poor results across all metrics. However, scaling down the UNet model (channels per level: (32, 64, 128) -> (16, 32), residual blocks per level: (2, 2, 2) -> (2, 2), embedding features: 64 ->16, attention head moved up one level) does not degrade performance appreciably.
>
> However, the inductive bias of the UNet architecture alone is unlikely to be the cause of our model’s strong performance. In our extended baseline comparison above we contrast our performance with a CLVM implementation that uses neural networks as encoders and decoders. After running an extensive hyperparameter sweep, we found that using MLPs over UNet-style convolutional encoders / decoders led to a substantial improvement in performance. If the UNet inductive bias was essential, it would be surprising to find the CLVM did not benefit from it. We added a summary of this discussion in Section 6.
> ### Linear Mixing and Known Mixing
> We agree with the reviewer that the principle limitation of our method is that it requires linear mixing. As it stands, our framework cannot easily be extended to the non-linear case without substituting our joint sampling for Gibbs sampling. It is likely possible to generalize our method further under a reasonable set of assumptions, but we cannot explore those extensions within the timeframe of the rebuttal. However, we think the linear assumption is a reasonable constraint for the current work given that our method is already more flexible than previous baseline methods. For example, we make no source assumptions, do not require contrastive observations, and can (theoretically) separate an arbitrary number of sources within a single observation.
>
> If we place no constraints on the mixing matrix, the problem becomes fully degenerate. Blind source separation methods must make strong assumptions about the sources in order to break this degeneracy [7]. However, it could be possible to incorporate a known prior distribution on the mixing matrix. One could then approximate each evaluation of Eq. 17 with a single Monte Carlo sample, and rely on the large dataset size and multiple training epochs to expose the model to the full mixing matrix distribution. Unfortunately, we were unable to run an experiment along these lines in the time allotted to the rebuttal.
>
> ### Additional notes
>
> * To ensure fair comparison, we ran a hyperparameter sweep for the baseline methods. Similarly, we made small architectural changes to improve the diffusion models used in our method and reran all experiments. Therefore, some of the metrics quoted in the table changed from our original submission.
>
> [1] Li, Didong, et al. "Probabilistic contrastive principal component analysis” (2020)
>
> [2] Severson, Kristen A., et al. “Unsupervised learning with contrastive latent variable models” (2019)
>
> [3] Heurtel-Depeiges, David, et al. "Listening to the noise: Blind denoising with Gibbs diffusion" (2024)
>
> [4] Lemos, Pablo, et al. "Pqmass: Probabilistic assessment of the quality of generative models using probability mass estimation" (2024)
>
> [5] Heusel, Martin, et al. "GANs trained by a two time-scale update rule converge to a local Nash equilibrium" (2017)
>
> [6] Abid, Abubakar, et al. “Contrastive Variational Autoencoder Enhances Salient Features” (2019)
>
> [7] Hyvärinen, Aapo, et al. "Independent component analysis: algorithms and applications" (2000)

---

> > ### Comment · Reviewer_G8fL · 2025-08-05
> >
> > I thank the authors for their detailed rebuttals to the reviews, and providing additional data and insights. Can you please clarify if you intend to include the new data and discussion points in the final version of the paper (or which of them)?

---

> > > ### Author Response · Authors · 2025-08-07
> > >
> > > Thank you for the follow-up. We have incorporated all of the new data and discussion points in the updated manuscript:
> > >
> > > Additional Results (Baselines and Metrics)
> > > - This table now replaces the original Tables 1 and 2 in the main text. Section 5 has been updated to highlight these new results.
> > >
> > > Runtime, Computational Cost, and Ablation Studies
> > > - These appear as tables in the supplementary material. We cite them in Section 5.2 and Section 6, and we have added a concise summary of their key takeaways in those sections.
> > >
> > > Linear / Known Mixing and Effectiveness
> > > - Section 6 now clarifies the mixing matrix limitations and discusses why we believe the method is effective for MVSS.

---

> > > > ### Comment · Reviewer_G8fL · 2025-08-07
> > > >
> > > > Thanks for clarifying, and thanks again for the additional data and insights. With these in the final paper, most of my  concerns, and those of other reviewers have been addressed. This is except for the request for results on other real-world datasets (beyond galaxy observations), which would be nice to have but I appreciate the time constraints. Based on this, I will raise my score to Accept.

---

### Official Review · Reviewer_BBqy · 2025-07-01

**Clarity:** 3
**Significance:** 2
**Originality:** 3
**Rating:** 4
**Confidence:** 3

**Summary:**

This paper proposes a method that should work under a general MVSS framework, where the sources are independent, the observations are noisy, and are incomplete and vary in resolution. The paper learns a diffusion model for each source and each of these separate diffusion models can be used to perform sampling and evaluate likelihoods. The main algorithm consists of an EM algorithm where the expectation step consists of sampling posterior using the reverse SDE and the maximization step consists of updating the model weights. For sampling, the authors propose an improved version of Gibbs sampling that reduces the sampling time. The experiments consist of using synthetic 1D manifold data, grassy MNIST dataset in which the objective is to separate the MNIST digits from the background, and astronomy images where the goal is to separate galaxies from random background light. The results show that the proposed method outperforms previous SOTA methods in accuracy and fidelity metrics.

**Questions:**

1 Are there some other relevant comparison methods that could be used to further confirm the efficacy of the method, such as more classical methods? It would also be useful to show runtimies of these methods, as in my experience diffusion model based methods tend to have a much slower runtime but produce higher quality images. It would be interesting to see the tradeoff between runtime and image quality.

2 Can any theoretical analysis be performed, either to bound the sampling error, show convergence toward distribution, or other theoretical guarantees?

3 In Table 2, the FID scores of the comparison methods seem extremely bad. Are there any weaknesses of those comparison methods that make them bad, and is there any way to adjust these or other existing methods to at least obtain reasonable results?

4 How does the method perform when certain parameters are adjusted? It would be useful to have some ablation studies that study the effects of individual components.

5 The paper assumes that the observations are linear combinations of the sources. Could the proposed methods be adapted in any way to handle cases where the observations are nonlinear combinations? Similarly it appears that the noise is assumed to be Gaussian in the experiments, but could the method be applied if the noise was Poisson, like it is in many real world applications?

Overall, the paper proposes an interesting method that shows promising results on a relevant problem and I am happy to revise my score should the concerns be addressed.

**Ethical Concerns:**

["NO or VERY MINOR ethics concerns only"]

**Final Justification:**

After reading the rebuttal, considering the additional experiments and results, and reading the other reviewer comments, I feel most of my concerns have been addressed so I increased my score.

**Limitations:**

yes

**Quality:**

3

**Strengths And Weaknesses:**

Strengths: The proposed method appears to be fairly flexible and can handle noisy, incomplete, and varying resolution data. Leveraging diffusion models which are known to learn priors very well, the paper describes how to combine them with EM to provide a data-driven solution to the problem. The paper demonstrates visually and quantitatively how the method is applied to real world problems in different fields, and the writing is clear and easy to follow.

Weaknesses: While the experiments performed and their corresponding settings are diverse, they are fairly sparse in detail. In particular, the tables of experimental results are small: more comparison methods could be included, including with more classical and non-deep learning methods. Furthermore, a greater range of experimental parameters for each setting could be observed: for example, different noise levels, image resolutions, etc. Also, a lack of ablation studies makes it difficult to determine whether each component of the model and algorithm helped improve the results significantly, or whether careful hyperparameter tuning is required to achieve stable results. Finally, there is a lack of theoretical analysis for the proposed algorithms.

---

> ### Author Rebuttal · Authors · 2025-07-31
>
> We would like to thank the reviewer for their detailed feedback. Below we address the concerns and questions brought up within the review:
>
> ### Additional Comparisons
>
> We agree that the work would benefit from more baselines, metrics, and a fuller exploration of the baselines. To address this we ran three baselines, PCPCA, CLVM-Linear, and CLVM-VAE, on all the contrastive experiments. To our knowledge, these are the best baselines available. We have run the Gibbs sampling baseline where computationally feasible. Where applicable, we track PSNR, the peak signal-to-noise ratio, PQM, the pqmass metric introduced in [1], FID, the FID score [2] using our MNIST classifier, and SD, the Sinkhorn divergence.
>
> |||Posterior|||||Prior||
> |-|:-:|:-:|:-:|:-:|:-:|:-:|:-:|:-:|
> |**Dataset / Method**|**PQM ↑**|**FID ↓**|**PSNR ↑**|**SD ↓**||**PQM ↑**|**FID ↓**|**SD ↓**|
> |**1D Manifold: Cont. 2 Sources**|||||||||
> |PCPCA [3]|0.0|--|9.35|7.69|¦|0.0|--|7.91|
> |CLVM - Linear [4]|0.0|--|9.58|5.80|¦|0.0|--|5.86|
> |CLVM - VAE [4]|0.0|--|17.15|1.81|¦|0.0|--|2.91|
> |DDPRISM-Gibbs [5]|0.0|--|12.66|3.96|¦|0.0|--|3.92|
> |DDPRISM-Joint [Ours]|**0.26**|--|**38.27**|**0.35**|¦|**0.01**|--|**0.37**|
> |**1D Manifold: Cont. 3 Sources**||||||||||
> |PCPCA [3]|0.0|--|6.89|12.57|¦|0.0|--|10.22|
> |CLVM - Linear [4]|0.0|--|11.64|2.03|¦|0.0|--|2.16|
> |CLVM - VAE [4]|0.0|--|13.09|2.22|¦|0.0|--|1.82|
> |DDPRISM-Gibbs [4]|0.0|--|9.50|4.50|¦|0.0|--|4.53|
> |DDPRISM-Joint [Ours]|0.0|--|**19.78**|**0.75**|¦|0.0|--|**0.78**|
> |**1D Manifold: Mix. ($f_{mix}=0.1$)**|||||||||
> |DDPRISM-Gibbs [5]|0.0|--|15.48|2.52|¦|0.0|--|2.43|
> |DDPRISM-Joint [Ours]|**0.001**|--|**24.15**|**0.05**|¦|0.0|--|**0.04**|
> |**GMNIST: Cont. Full Resolution**|||||||||
> |PCPCA [3]|0.0|22.30|18.99|--|¦|0.0|176.0|--|
> |CLVM - Linear [4]|0.0|101.3|13.30|--|¦|0.0|139.9|--|
> |CLVM - VAE [4]|0.0|18.87|14.56|--|¦|0.0|57.67|--|
> |DDPRISM-Joint [Ours]|**1.00**|**1.57**|**25.60**|--|¦|**0.20**|**20.10**|--|
> |**GMNIST: Cont. Downsampled**|||||||||
> |PCPCA [3]|0.0|121.7|14.08|--|¦|0.0|115.4|--|
> |CLVM - Linear [4]|0.0|199.5|12.16|--|¦|0.0|211.4|--|
> |CLVM - VAE [4]|0.0|1008.0|8.48|--|¦|0.0|737.0|--|
> |DDPRISM-Joint [Ours]|0.0|**60.34**|**17.09**|--|¦|0.0|**79.43**|--|
>
> To ensure fair comparison, we ran an extensive hyperparameter sweep for the baseline methods. Similarly, we made small architectural changes to improve the diffusion models used in our method and reran all experiments. We did not run hyperparameter sweeps on our method due to time constraints for the rebuttal, but will do so for the camera-ready paper. Note that some of the metrics quoted in the original submission have changed, including improvements in the PCPCA performance that concerned the reviewer. However, there remains a substantial gap between the performance of the baselines and our model.
>
> ### Runtime and Computational Cost
>
> We provide a detailed comparison of the computational cost for each method in the table below:
>
> |||Training Time||¦||Inference Time per Sample||
> |-|:-:|:-:|:-:|:-:|:-:|:-:|:-:|
> |**Method**|**1D Manifold**|**GMNIST Full Res.**|**Galaxy Images**|¦|**1D Manifold**|**GMNIST Full Res.**|**Galaxy Images**|
> |PCPCA [3]|5s|1m|--|¦|<0.1ms|1.5ms|--|
> |CLVM - Linear [4]|1.5m|10m|--|¦|<0.1ms|1ms|--|
> |CLVM - VAE [4]|18m|33m|--|¦|<0.1ms|1ms|--|
> |DDPRISM-Joint [Ours]|32h|68h|48h|¦| 22ms| 90ms|1.5s|
>
> All timing was done on NVIDIA A100 GPUs. Our method is more computationally expensive than the baselines, almost entirely due to the cost of sampling from the diffusion model. However, this computational cost comes with significant improvements in performance and sample quality. In our ablation studies below we also compare the performance as a function of EM laps which can be used to control the tradeoff between training time and sample quality for our method.
>
> We added a section to the appendix that includes the full timing comparison above and point to that section in our discussion and limitations section and in the results for each experiment.
>
> ### Ablation Studies
>
> Per the reviewer’s suggestion, in order to clarify the importance of individual components of our method, we conducted an ablation study. We explored the effect of the following components on the performance of the method: (1) diffusion model architecture; (2) training length (in terms of the number of EM iterations), (3) initialization strategy, and (4) dataset size:
> |||Posterior|||||Prior||
> |-|:-:|:-:|:-:|:-:|:-:|:-:|:-:|:-:|
> |**GMNIST Full Res.**|**PQM ↑**|**FID ↓**|**PSNR ↑**|**SD ↓**||**PQM ↑**|**FID ↓**|**SD ↓**|
> |**Model Architecture**|||||||||
> |MLP|0.05|49.03|17.93|--|¦|0.|215.36|--|
> |UNet, small|1.00|2.47|25.28|--|¦|0.06|**15.38**|--|
> |UNet, default|**1.00**|**1.57**|**25.60**|--|¦|**0.20**|20.10|--|
> |**Training Length (EM laps)**|||||||||
> |2|0.|96.85|16.62|--|¦|0.|199.70|--|
> |8|**1.00**|**0.04**|**27.15**|--|¦|0.14|**17.35**|--|
> |32|1.00|2.26|25.66|--|¦|0.08|27.96|--|
> |64 (Default)|1.00|1.57|25.60|--|¦|**0.20**|20.10|--|
> |**Initialization (# of Gaussian EM laps)** |||||||||
> |0 (Random)|0.97|10.41|23.35|--|¦|0.01|22.22|--|
> |4|**1.00**|**0.00**|26.80|--|¦|0.15|24.33|--|
> |16|**1.00**|**0.00**|**27.02**|--|¦|**0.21**|**6.31**|--|
> |64 (Default)|**1.00**|1.57|25.60|--|¦|0.20|20.10|--|
> |**Dataset Size**|||||||||
> |Full Dataset|**1.00**|**1.57**|**25.60**|--|¦|**0.20**|20.10|--|
> |1/4th Dataset|1.00|4.64|23.67| -- |¦|0.14|21.36|--|
> |1/16th Dataset|0.99|10.19|20.97|--|¦|0.07|**15.16**|--|
> |1/64th Dataset|0.0|38.34|15.34|--|¦|0.0|36.55|--|
>
> We find that model architecture is important for performance: using an MLP-based architecture (5 hidden layers with 2048 hidden features per layer) gives poor results across all metrics. However, scaling down the UNet model (channels per level: (32,64,128)->(16,32), residual blocks per level: (2,2,2)->(2,2), embedding features: 64->16, attention head moved up one level) does not degrade performance appreciably.
>
> We also find that the model achieves good performance after as few as 8 EM iterations and that longer training leads to a slight degradation in performance. In the table comparing our performance to the baselines we report the results for the model trained for the default number of EM laps.
>
> For dataset size, we train our method using 1/4th, 1/16th, and 1/64th of the original grass and grass+MNIST datasets. The 1/16th and 1/64th runs are given 1/4th as many EM laps as the original training to account for the smaller dataset, but all other hyperparameters are unchanged. There is clear degradation in performance as the grass and MNIST datasets are reduced in size. However, even with 1/16th of the original dataset (2048 grass images, 512 MNIST digits) our method still outperforms the baselines run on the full dataset.
>
> Surprisingly, the number of Gaussian EM laps used to generate the initial samples (and train the initial diffusion model) has minimal impact on performance. Only starting from a randomly initialized model considerably reduces performance.
>
> We added these ablation studies to the appendix, referred to them in Section 5.2, and discussed them in Section 6. If the reviewer would like additional ablations, we can incorporate them in the camera-ready version.
>
> ### Linear Mixing and Gaussian Noise Assumptions
>
> As it stands, our framework cannot easily be extended to the non-linear case without substituting our joint sampling for Gibbs sampling. It is likely possible to generalize our method further under a reasonable set of assumptions about the form of the mapping (i.e. a small perturbative expansion around linear), but we leave this for future work. We think the linear assumption is a reasonable constraint for the current paper given that our method is already more flexible than previous baseline methods.
>
> The Gaussian noise assumption, however, can be relaxed within our framework. It would require access to “noise-only” views or the ability to sample from a known noise distribution. Essentially, one can consider the noise as an additional source and pre-train its corresponding diffusion model with the noise samples. From there, the rest of our machinery operates identically to the experiments presented in the paper, with the exception that the weights of the noise diffusion model are never updated. The only limitation is that the linear mixing assumption must be preserved. That is, noise that depends on the signal (for example, Poisson noise mentioned by the reviewer) cannot be modeled in this way.
>
> ### Theoretical Bounds
>
> To our knowledge, there are three relevant convergence guarantees for this method. First, in the limit of perfect optimization, infinite training examples, and a sufficiently expressive architecture, the score-based diffusion model is guaranteed to converge to the correct score [6,7]. In the limit of infinite sampling steps, our sampling method is guaranteed to sample from the learned prior [8,9]. Finally, substituting the prior score for the posterior score in Equation 6 is guaranteed to return posterior samples. However, we do not meet the requirements for any of these guarantees due to finite training dataset, finite sampling steps, and limitations of the MMPS approximation. We are unfortunately not aware of any relevant convergence bounds.
>
> [1] Lemos, Pablo, et al. (2024)
> [2] Heusel, Martin, et al. (2017)
> [3] Li, Didong, et al. (2020)
> [4] Severson, Kristen A., et al. (2019)
> [5] Heurtel-Depeiges, David, et al. (2024)
> [6] Hyvärinen, A. (2005)
> [7] Song, Yang et al. (2019)
> [8] Sohl-Dickstein, Jascha, et al. (2015)
> [9] Song, Yang, et al. (2020)

---

> > ### Author Response · Authors · 2025-08-07
> >
> > We ask the reviewer to please let us know if they have any remaining questions. We want to check in as the response period deadline is approaching.

---

> > > ### Comment · Reviewer_BBqy · 2025-08-08
> > >
> > > After reading the rebuttal, considering the additional experiments and results, and reading the other reviewer comments, I feel most of my concerns have been addressed so I increased my score.

---

### Official Review · Reviewer_zJxX · 2025-07-05

**Clarity:** 4
**Significance:** 3
**Originality:** 3
**Rating:** 5
**Confidence:** 1

**Summary:**

This paper proposes a diffusion model-based approach for multi-view source separation, where multiple noisy observations contain different linear mixtures of unknown sources. Unlike prior methods relying on contrastive setups or simple priors, the authors use diffusion models to directly learn source priors and perform posterior sampling. Integrated into an EM framework, the method iteratively refines source estimates without assuming isolated source observations. Experiments on synthetic and real-world data show strong performance over existing MVSS methods, particularly in noisy, incomplete, and heterogeneous settings.

**Questions:**

- The proposed EM framework with diffusion priors is computationally expensive, especially in posterior sampling. Can you provide quantitative comparisons of runtime and memory usage against baselines? How feasible is your method for larger-scale problems (like high-resolution astronomy datasets)?

- The method currently assumes linear mixing and additive Gaussian noise. Can your framework be extended to nonlinear observation models, or do you plan to explore this in the future?

- Diffusion models are known to be data-hungry. How does your method perform when the number of observations is small? Could you provide experiments in low-data regimes or discuss ways to mitigate overfitting in such cases?

**Ethical Concerns:**

["NO or VERY MINOR ethics concerns only"]

**Final Justification:**

The authors have satisfactorily addressed my pre-rebuttal concerns on computational cost by providing detailed runtime and memory comparisons against baselines, and clarifying feasibility for large-scale datasets.They have conducted additional experiments to assess performance under limited data, showing graceful degradation and still outperforming baselines in most settings, which resolves my concern. The limitation regarding linear mixing and Gaussian noise assumptions remains, but the authors provided a clear discussion of possible extensions and future directions.

Given the strong empirical results, thorough rebuttal clarifications, I maintain my positive assessment and recommendation for acceptance.

**Limitations:**

Yes.

**Paper Formatting Concerns:**

None.

**Quality:**

4

**Strengths And Weaknesses:**

Strengths:
- The proposed framework is general. The paper proposes a general MVSS solution applicable to various scientific domains, without relying on contrastive assumptions or isolated source data. The use of diffusion priors is also novel, the paper introduces diffusion models as flexible source priors for source separation, enabling expressive modeling of complex distributions.

- The paper presents strong results in challenging settings, which demonstrates state-of-the-art results on both synthetic and real-world datasets, especially under noisy and incomplete data.

- The paper has clear theoretical foundation, the method is grounded in a principled EM framework with posterior sampling, and provides solid mathematical formulation.

- The results show that the approach working across different problem scales, from low-dimensional manifolds to real astronomical images.

Weaknesses:
- The EM iterations and diffusion-based sampling are computationally expensive, making scalability to large datasets questionable.

- The work requires large datasets to train the diffusion priors effectively; unclear how well it performs in low-data regimes.

---

> ### Author Rebuttal · Authors · 2025-07-31
>
> We would like to thank the reviewer for their thoughtful feedback. Below we address the concerns and questions brought up within the review.
>
> ### Runtime and Computational Cost
>
> We provide a detailed comparison of the computational time for each method in the table below:
>
> |||Training Time||¦||Inference Time per Sample||
> |-|:-:|:-:|:-:|:-:|:-:|:-:|:-:|
> |**Method**|**1D Manifold**|**GMNIST Full Res.**|**Galaxy Images**|¦|**1D Manifold**|**GMNIST Full Res.**|**Galaxy Images**|
> |PCPCA [1]|5s|1m|--|¦|<0.1ms|1.5ms|--|
> |CLVM - Linear [2]|1.5m|10m|--|¦|<0.1ms|1ms|--|
> |CLVM - VAE [2]|18m|33m|--|¦|<0.1ms|1ms|--|
> |DDPRISM-Joint [Ours]|32h|68h|48h|¦| 22ms| 90ms|1.5s|
>
> CLVM-Linear denotes a contrastive latent-variable model with a linear transformation from observation to latents, and CLVM-VAE replaces the linear model with a neural network encoder / decoder. All timing was done on NVIDIA A100 GPUs. As the reviewer notes, our method is more computationally expensive than the baselines, almost entirely due to the cost of sampling from the diffusion model (since sampling must be done both during the EM loop and at inference time). However, this computational cost comes with significant improvements in performance. Below is the comparison of our model to the baselines across several experiments and metrics:
>
> |||Posterior|||||Prior||
> |-|:-:|:-:|:-:|:-:|:-:|:-:|:-:|:-:|
> |**Dataset / Method**|**PQM ↑**|**FID ↓**|**PSNR ↑**|**SD ↓**||**PQM ↑**|**FID ↓**|**SD ↓**|
> |**1D Manifold: Cont. 2 Sources**|||||||||
> |PCPCA [1]|0.0|--|9.35|7.69|¦|0.0|--|7.91|
> |CLVM - Linear [2]|0.0|--|9.58|5.80|¦|0.0|--|5.86|
> |CLVM - VAE [2]|0.0|--|17.15|1.81|¦|0.0|--|2.91|
> |DDPRISM-Gibbs [3]|0.0|--|12.66|3.96|¦|0.0|--|3.92|
> |DDPRISM-Joint [Ours]|**0.26**|--|**38.27**|**0.35**|¦|**0.01**|--|**0.37**|
> |**1D Manifold: Cont. 3 Sources**||||||||||
> |PCPCA [1]|0.0|--|6.89|12.57|¦|0.0|--|10.22|
> |CLVM - Linear [2]|0.0|--|11.64|2.03|¦|0.0|--|2.16|
> |CLVM - VAE [2]|0.0|--|13.09|2.22|¦|0.0|--|1.82|
> |DDPRISM-Gibbs [3]|0.0|--|9.50|4.50|¦|0.0|--|4.53|
> |DDPRISM-Joint [Ours]|0.0|--|**19.78**|**0.75**|¦|0.0|--|**0.78**|
> |**1D Manifold: Mix. ($f_{mix}=0.1$)**|||||||||
> |DDPRISM-Gibbs [3]|0.0|--|15.48|2.52|¦|0.0|--|2.43|
> |DDPRISM-Joint [Ours]|**0.001**|--|**24.15**|**0.05**|¦|0.0|--|**0.04**|
> |**GMNIST: Cont. Full Resolution**|||||||||
> |PCPCA [1]|0.0|22.30|18.99|--|¦|0.0|176.0|--|
> |CLVM - Linear [2]|0.0|101.3|13.30|--|¦|0.0|139.9|--|
> |CLVM - VAE [2]|0.0|18.87|14.56|--|¦|0.0|57.67|--|
> |DDPRISM-Joint [Ours]|**1.00**|**1.57**|**25.60**|--|¦|**0.20**|**20.10**|--|
> |**GMNIST: Cont. Downsampled**|||||||||
> |PCPCA [1]|0.0|121.7|14.08|--|¦|0.0|115.4|--|
> |CLVM - Linear [2]|0.0|199.5|12.16|--|¦|0.0|211.4|--|
> |CLVM - VAE [2]|0.0|1008.0|8.48|--|¦|0.0|737.0|--|
> |DDPRISM-Joint [Ours]|0.0|**60.34**|**17.09**|--|¦|0.0|**79.43**|--|
>
> PSNR is the peak signal-to-noise ratio, PQM is the pqmass metric introduced in [4], FID is the FID score [5] using our MNIST classifier, and SD is the sinkhorn divergence. Note that we’ve replaced the CVAE [6] baseline with CLVM-VAE [2] since the latter is a superset of the former and achieves better performance. The substantive gap between our method and the baselines justifies the additional computational resources required to train and sample the diffusion model.
>
> In terms of the feasibility for large datasets, the galaxy images experiment serves as a good baseline. The HST images we use are much higher resolution that ground-based observations [7,8] and equivalent in resolution to other current space missions [9]. Training the random and galaxy images took 48 GPU hours, and sampling from the posterior of an individual observation takes 1.5 seconds. For the ~80k galaxy images in the catalog, this amounts to under 34 GPU hours to draw one posterior sample for each galaxy. This compute budget is realistic for an academic lab.
>
> We have added a section to the appendix that includes the full timing comparison above and point to that section in our discussion and limitations section and in the results for each experiment.
>
> ### Performance as a Function of Dataset Size
>
> We agree with the reviewer that, by using diffusion models to fit our prior distributions, our method will be sensitive to dataset size and likely perform poorly in the low data regime. Some work has explored solutions to reduce dataset size requirements, including data augmentation [11], patch-wise diffusion training [12], latent-space diffusion models [10, 13], and transfer learning [14]. It would be possible to incorporate these methods during our maximization step.
>
> To better quantify our dependence on the dataset size, we ran a set of ablation studies on our performance on the full-resolution Grassy MNIST experiment. We train our method using 1/4th, 1/16th, and 1/64th of the original grass and grass+MNIST datasets. The 1/16th and 1/64th runs are given 1/4th as many EM laps as the original training to account for the smaller dataset, but all other hyperparameters are unchanged:
>
> |||Posterior|||||Prior||
> |-|:-:|:-:|:-:|:-:|:-:|:-:|:-:|:-:|
> |**GMNIST Full Res.** |**PQM ↑**|**FID ↓**|**PSNR ↑**|**SD ↓**| |**PQM ↑**|**FID ↓**|**SD ↓**|
> |Full Dataset|**1.00**|**1.57**|**25.60**|--|¦|**0.20**|20.10|--|
> |1/4th Dataset|1.00|4.64|23.67| -- |¦|0.14|21.36|--|
> |1/16th Dataset|0.99|10.19|20.97|--|¦|0.07|**15.16**|--|
> |1/64th Dataset|0.0|38.34|15.34|--|¦|0.0|36.55|--|
>
> As the reviewer suggested, there is clear degradation in performance as the grass and MNIST datasets are reduced in size. However, even with 1/16th of the original dataset (2048 grass images, 512 MNIST digits) our method still outperforms the baselines run on the full dataset. We have added the dataset scaling study to the appendix and referenced it in the Discussion and Limitations section. We have also added a brief discussion of strategies that can be used to improve the performance of diffusion models in the low-data regime.
>
> ### Linear Mixing and Gaussian Noise Assumptions
>
> The reviewer is correct that our current method assumes both a linear observation model and Gaussian noise. As it stands, our framework cannot easily be extended to the non-linear case without substituting our joint sampling for Gibbs sampling. It is likely possible to generalize our method further under a reasonable set of assumptions, but we reserve that for future work.
>
> The Gaussian noise assumption can be relaxed within our framework. It would require access to “noise-only” views or the ability to sample from a known noise distribution. One can consider the noise as an additional source and pre-train its corresponding diffusion model with the noise samples. From there, the rest of our machinery operates identically to the experiments presented in the paper, with the weights of the noise diffusion model frozen. However, the linear mixing assumption must be preserved. Noise that depends on the signal (for example Poisson noise from photon counts) cannot be modeled in this way. We leave the extension to non-linear noise models for future work.
>
> ### Additional notes
>
> * For the posterior metrics, we are comparing posterior samples to the original signal, meaning that they are not independent draws. Therefore, it is possible to get large positive pqmass values and large negative sinkhorn divergence values. The posterior metrics are meant to compare the reconstruction produced by different methods,  and should not be interpreted in absolute terms. We have added a sentence to the main body of the paper to clarify this.
> * To ensure fair comparison, we ran a hyperparameter sweep for the baseline methods. Similarly, we made small architectural changes to improve the diffusion models used in our method and reran all experiments. Therefore, some of the metrics quoted in the table changed from our original submission.
>
> [1] Li, Didong, et al. "Probabilistic contrastive principal component analysis” (2020)
> [2] Severson, Kristen A., et al. “Unsupervised learning with contrastive latent variable models” (2019)
> [3] Heurtel-Depeiges, David, et al. "Listening to the noise: Blind denoising with Gibbs diffusion" (2024)
> [4] Lemos, Pablo, et al. "Pqmass: Probabilistic assessment of the quality of generative models using probability mass estimation" (2024)
> [5] Heusel, Martin, et al. "GANs trained by a two time-scale update rule converge to a local Nash equilibrium" (2017)
> [6] Abid, Abubakar, et al. “Contrastive Variational Autoencoder Enhances Salient Features” (2019)
> [7] Koekemoer, Anton M., et al. "The COSMOS survey: Hubble space telescope advanced camera for surveys observations and data processing" (2007)
> [8] Ivezić, Željko, et al. "LSST: from science drivers to reference design and anticipated data products" (2019)
> [9] Gardner, Jonathan P., et al. “The James Webb Space Telescope Mission” (2023).
> [10] Rombach, Robin, et al. "High-resolution image synthesis with latent diffusion models" (2022)
> [11] Karras, Tero, et al. "Elucidating the design space of diffusion-based generative models" (2022)
> [12] Wang, Zhendong, et al. "Patch diffusion: Faster and more data-efficient training of diffusion models" (2023)
> [13] Zhang, Zhaoyu, et al. “Training Diffusion-Based Generative Models with Limited Data” (2025)
> [14] Hur, Jiwan, et al. "Expanding expressiveness of diffusion models with limited data via self-distillation based fine-tuning" (2024)

---

> > ### Author Response · Authors · 2025-08-07
> >
> > We ask the reviewer to please let us know if they have any remaining questions. We want to check in as the response period deadline is approaching.

---

> > ### Comment · Reviewer_zJxX · 2025-08-07
> >
> > Thank you for your thoughtful and thorough rebuttal. It has resolved the majority of my concerns, and I appreciate the results and clarifications the authors provided. I will keep my positive score.

---

### Decision · Program_Chairs · 2025-09-17

**Decision:**

Accept (poster)

**Comment:**

The paper received uniformly positive reviews with scores of 4, 5, 5, 5 (mean: 4.75). Reviewers praised the novelty of introducing diffusion priors for multi-view source separation (zJxX, BBqy, G8fL, FLez), the principled EM-based framework (zJxX, G8fL), strong empirical results across both synthetic and real-world datasets (zJxX, BBqy, G8fL), and clarity of presentation (BBqy, FLez). Initial concerns on scalability, evaluation breadth, ablations, and initialization were effectively addressed in the rebuttal with additional experiments, runtime analyses, and expanded discussion of limitations.

After thoroughly reviewing the paper, the reviews, and the authors’ rebuttal, the AC concurs with the reviewers’ highly positive consensus and therefore recommends the paper for **acceptance**.

For the camera-ready version, the authors should ensure all clarifications and additional results from the rebuttal are incorporated into the main paper and supplementary material. In particular:

**1. Evaluation (BBqy, G8fL):** Integrate the extended baseline comparisons and expanded metrics in the rebuttal directly into the main text.

**2. Ablations and initialization (FLez, BBqy, G8fL):** Summarize the ablation results on architecture, EM iterations, initialization strategies, and dataset size to clarify the robustness of the approach.

**3. Computational cost (zJxX, G8fL, BBqy):** Include the runtime and memory comparisons with baselines, along with the discussion of scalability to large-scale datasets (e.g., astronomy).

**4. Limitations and scope (zJxX, G8fL):** Retain the discussion on linear mixing and Gaussian noise assumptions, and highlight possible extensions or constraints explicitly in the conclusion.